# Genomic insights into the evolution of *Echinochloa* species as weed and orphan crop

Dongya Wu [1,18], Enhui Shen[1,2,18], Bowen Jiang[1], Yu Feng[3,4], Wei Tang[5], Sangting Lao[1], Lei Jia[1], Han-Yang Lin[3], Lingjuan Xie[1], Xifang Weng[1], Chenfeng Dong[1], Qinghong Qian[1], Feng Lin[1], Haiming Xu [1], Huabing Lu[6], Luan Cutti [7], Huajun Chen[8], Shuiguang Deng[8], Longbiao Guo [5], Tse-Seng Chuah[9], Beng-Kah Song[10], Laura Scarabel[11], Jie Qiu [12], Qian-Hao Zhu[13], Qin Yu [14], Michael P. Timko[15], Hirofumi Yamaguchi[16], Aldo Merotto Jr[7], Yingxiong Qiu[3], Kenneth M. Olsen [17], Longjiang Fan[1,2] & Chu-Yu Ye [1✉]

As one of the great survivors of the plant kingdom, barnyard grasses (*Echinochloa* spp.) are the most noxious and common weeds in paddy ecosystems. Meanwhile, at least two *Echinochloa* species have been domesticated and cultivated as millets. In order to better understand the genomic forces driving the evolution of *Echinochloa* species toward weed and crop characteristics, we assemble genomes of three *Echinochloa* species (allohexaploid *E. crus-galli* and *E. colona*, and allotetraploid *E. oryzicola*) and re-sequence 737 accessions of barnyard grasses and millets from 16 rice-producing countries. Phylogenomic and comparative genomic analyses reveal the complex and reticulate evolution in the speciation of *Echinochloa* polyploids and provide evidence of constrained disease-related gene copy numbers in *Echinochloa*. A population-level investigation uncovers deep population differentiation for local adaptation, multiple target-site herbicide resistance mutations of barnyard grasses, and limited domestication of barnyard millets. Our results provide genomic insights into the dual roles of *Echinochloa* species as weeds and crops as well as essential resources for studying plant polyploidization, adaptation, precision weed control and millet improvements.

[1] Institute of Crop Science & Institute of Bioinformatics, Zhejiang University, Hangzhou 310058, China. [2] Zhejiang University Zhongyuan Institute, Zhengzhou 450000, China. [3] Institute of Ecology, Zhejiang University, Hangzhou 310058, China. [4] Chengdu Institute of Biology, Chinese Academy of Sciences, Chengdu 610041, China. [5] State Key Laboratory of Rice Biology, China National Rice Research Institute, Hangzhou 310006, China. [6] Institute of Maize and Upland Grain, Zhejiang Academy of Agricultural Sciences, Dongyang 322105, China. [7] Department of Crop Sciences, Agricultural School, Federal University of Rio Grande do Sul, Porto Alegre, RS 91540-000, Brazil. [8] College of Computer Science and Technology, Zhejiang University, Hangzhou 310058, China. [9] Faculty of Plantation and Agrotechnology, Universiti Teknologi MARA, 02600 Arau, Perlis, Malaysia. [10] School of Science, Monash University Malaysia, 46150 Bandar Sunway, Selangor, Malaysia. [11] Istituto per la Protezione Sostenibile delle Piante (IPSP), CNR, Viale dell'Università, 16, 35020 Legnaro (PD), Italy. [12] Shanghai Key Laboratory of Plant Molecular Sciences, College of Life Sciences, Shanghai Normal University, Shanghai 200235, China. [13] CSIRO Agriculture and Food, GPO Box 1700 Canberra, ACT 2601, Australia. [14] Australian Herbicide Resistance Initiative, School of Agriculture and Environment, University of Western Australia, Crawley, WA 6009, Australia. [15] Department of Biology, University of Virginia, Charlottesville, VA 22904, USA. [16] Osaka Prefecture University, Sakai, Osaka 599-8531, Japan. [17] Department of Biology, Washington University in St. Louis, St. Louis, MO 63130, USA. [18] These authors contributed equally: Dongya Wu, Enhui Shen. ✉email: yecy@zju.edu.cn

Weeds are among the super survivors of the plant kingdom as they generally have large populations and wide geographic distributions[1]. Barnyard grasses (*Echinochloa* P. Beauv.) are among the most problematic and dominant weeds in world agricultural systems, especially in rice paddy fields[2]. *Echinochloa* species are distributed globally from tropical to temperate regions and across a wide range of habitats, including agricultural and non-agricultural fields. Barnyard grasses in agroecosystems exhibit typical weed adaptation syndromes, such as aggressive competition with high nitrogen utilization, crop mimicry, allelopathy, and abiotic stress resistance (including herbicide resistance)[3–7]. The biosynthesis and release of allelochemicals (e.g., benzoxazinoid and momilactone) allow barnyard grasses to suppress the growth of surrounding plants (including crops) and to resist pathogen attack[8,9]. The tillering and grain yield of rice can be reduced by up to 50–70% because of the competition pressure of barnyard grasses[5,10]. Herbicides are widely used for the control of barnyard grasses, which has resulted in resistance evolution to herbicides of multiple sites-of-action in many barnyard grass populations across the world[11–14]. These mainly include resistance to the auxin mimic quinclorac and inhibitors of acetolactate synthase (ALS) (e.g., penoxsulam and bispyribac-sodium), acetyl-CoA carboxylase (ACCase) (e.g., fenoxaprop-ethyl and cyhaloprop-butyl), 5-enolpyruvylshikimate-3-phosphate synthase (EPSPS) (e.g., glyphosate), photosystem II (PSII; e.g., atrazine and propanil), and very-long-chain fatty acid synthase (e.g., butachlor)[13]. Global resistance evolution to major herbicides in barnyard grass populations threatens the sustainability of herbicide technology and, thus, of rice production. Overall, barnyard grasses are excellent resources or systems for understanding plant tolerance to stress, due to their high adaptability to artificial and wild environments.

Owing to their highly plastic plant architecture and morphological similarities across the genus, barnyard grass species are very difficult to delimit on the basis of morphological criteria. Consequently, species names are often used indiscriminately without standard identification criteria, leading to confusion in global species distribution patterns and impeding local precision weed control. Among ~50 species of the genus *Echinochloa*[15], *E. crus-galli* (L.) P. Beauv. (2n = 6× = 54), *E. colona* (L.) Link (2n = 6× = 54), and *E. oryzicola* Vasinger (2n = 4× = 36) [synonym *E. phyllopogon* (Stapf) Stapf ex Kossenko] are the three most widespread species in agroecosystems[2]. *Echinochloa oryzicola* is assumed to be an ancestral parent of *E. crus-galli*[16].

At least two barnyard millet species have been domesticated and cultivated as crops[17], *E. crus-galli* var. esculenta [synonym *E. utilis* Ohwi & Yabuno or *E. esculenta* (A. Braun) H. Scholz] and *E. colona* var. *frumentacea* (Link) Ridl. (synonym *E. frumentacea* Link)[7,18–20] (Supplementary Note 1). The domestication of *E. crus-galli* var. esculenta has been estimated to have occurred ~5000 years ago, and it is cultivated mainly in the cool and dry areas of East Asia where rice does not grow well[21]. *Echinochloa colona* var. *frumentacea* is cultivated as a cereal crop called 'jungle rice', mainly in the Indian subcontinent[7,18]. Despite the good nutritional and agronomic value of barnyard millets[7], little is known about the processes and genetic basis of their domestication.

To date, only the genome of the diploid species *E. haploclada* (Stapf) Stapf (2n = 2× = 18), which is a closely related species of maternal donor of *E. crus-galli*, has been sequenced and assembled to the chromosome level in the *Echinochloa* genus[10]. To better understand the evolution and adaptation of *Echinochloa* as weeds and to gain insights into the domestication of barnyard millets as crops, we sequence and assemble the genomes of three prevalent *Echinochloa* species (*E. crus-galli*, *E. colona*, and *E. oryzicola*) using diploid-assisted scaffolding method DipHic and perform a large-scale genomic investigation using a global panel of 737 barnyard grasses and millets from 16 countries covering rice production areas worldwide. In addition, we assess the genomic characteristics of the *Echinochloa* genus that allows its species to evolve as problematic weeds and orphan crops.

## Results

**Assembly of *Echinochloa* reference genomes.** We sequenced one domesticated millet accession (*E. colona* var. *frumentacea*) and two weedy accessions (*E. crus-galli* and *E. oryzicola*) to generate genomes for the three *Echinochloa* species.

*Echinochloa colona* is an allohexaploid species, and the domesticated millet accession *E. colona* var. *frumentacea*, PI463783, was selected as a candidate representative of the *E. colona* genome. Var. *frumentacea* is cross-compatible with other varieties, such as the weedy var. *colona*[21]. The estimated genome size of var. *frumentacea* (PI463783), based on the *k*-mer survey using Illumina short reads, was ~1.18 Gb, which was consistent with its C-value of 1.30 pg/1C[22]. Its heterozygosity level was estimated to be 0.131%. Using 48× coverage (57.2 Gb) of PacBio HiFi reads with an average read length of 15.1 kb, a de novo assembly yielded a genome of 1.13 Gb with a contig N50 size of 24.8 Mb. The assembly was further refined using chromosome contact maps with ~100× Hi-C data. Owing to the nature of hexaploidy, the sequence similarity among the three subgenomes severely reduced the accuracy of the chromosome scaffolding using regular Hi-C-assisted methods (Supplementary Fig. 1a). Therefore, we developed a diploid-assisted scaffolding method (named as DipHiC) to scaffold the genomes of polyploid *Echinochloa* species (Fig. 1a; See details at Supplementary Note 2). DipHiC includes two steps, subgenome distinguishing and chromosome building. To assign contigs to the three subgenomes of *E. colona*, we integrated the genetic relationship to diploid species, allelic contig information, and inter-contig interactions via multiple rounds of clustering and correction (Supplementary Fig. 2). Consequently, 97.1% of contig sequences were assigned into the three subgenomes, which were designated DH, EH, and FH (AH, BH, and CH have been designated for the three subgenomes of *E. crus-galli*, with 'H' referring to hexaploid) (Supplementary Note 2). In each subgenome, the directional interactions and chromosome pre-assignment with the diploid reference (*E. haploclada*) were utilized to construct linkages among contigs. After correction and ordering, we finally assembled the genome of *E. colona* with subgenome sizes of 436.4 (DH), 310.5 (EH), and 347.1 Mb (FH) (Fig. 1b; Table 1; Supplementary Fig. 1b).

The quality of the assembly was validated through the mapping of 90.4% of the transcriptomic reads obtained by RNA-Seq and 97.9% of the genomic short reads obtained by Illumina sequencing to the assembly. In addition, 97.0% of the 4896 core conserved Poales genes (BUSCO) were completely recovered, and the long terminal repeat (LTR) assembly index (LAI) score was 22.5 (where > 20 indicates a gold quality). Clustering of the counts of 13-mer sequences that differentiate homoeologous chromosomes enables consistent partitioning of the genome into subgenomes (Supplementary Fig. 3). Centromeric regions could be found in all chromosomes containing a 155-bp tandem repeat, similar to other grass species[23], and telomeric regions could be determined in most chromosomes, including 10 two-end chromosomes and 15 single-end chromosomes. High genomic synteny was observed between the *E. colona* assembly and the diploid and outgroup genomes (*Setaria italica* and *Oryza sativa*) (Supplementary Fig. 4a). Cumulatively, these results underscore the reliability of the *E. colona* assembly.

The draft genomes (contig-level) of hexaploid *E. crus-galli* (STB08) and tetraploid *E. oryzicola* (ZJU2) have been assembled

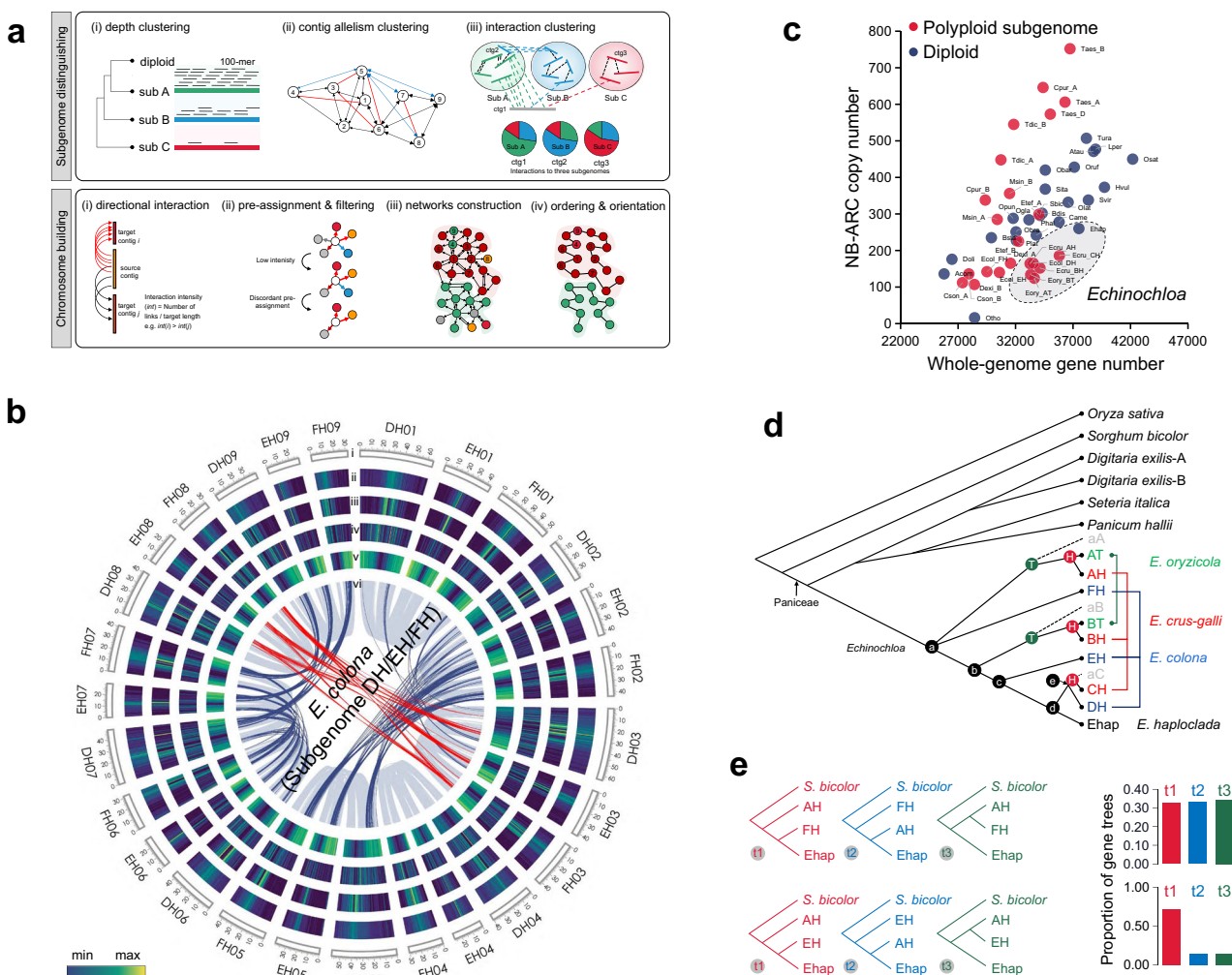

**Fig. 1 Assemblies of reference genomes and speciation of the _Echinochloa_ genus. a** Overview of the diploid-assisted scaffolding method (DipHiC) workflow in hexaploid assembly. Two modules, subgenome distinguishing and chromosome building, are implemented in DipHiC. Depth information from mapping the 100-mers generated from the diploid assembly to the hexaploid draft assembly, contig allelism, and inter-contig interactions are integrated in the subgenome distinguishing module. Directional interactions among contigs within single subgenome and chromosome group pre-assignment are determined using diploid assembly to anchor contigs and build chromosomes. **b** Features of the _Echinochloa colona_ reference genome. The chromosomes are shown in track i. Track ii–v show the densities of repeat elements (ii), _Gypsy_ (iii), _Copia_ (iv), and gene density (v). Track vi shows the syntenic blocks. Gray links show syntenic relationship among homologous subgenomes. Dark-blue and red links show syntenic relationships derived from an ancient whole-genome duplication event shared by Poaceae. **c** Distribution of the NB-ARC gene family size in different Poaceae genomes. See the full names of species in Supplementary Data 1. _Echinochloa_ (sub)genomes are highlighted in the dashed ellipse. **d** Phylogeny of _Echinochloa_ by integrating single-copy gene trees using ASTRAL. Dashed branches represent extinction or unknown status. 'T' and 'H' in the green and red nodes represent tetraploidization and hexaploidization, respectively. _D. exilis_-A and -B, subgenomes A and B in _Digitaria exilis_; Ehap, _Echinochloa haploclada_; AT and BT, subgenomes A and B in tetraploid _E. oryzicola_; AH, BH, and CH, subgenomes A, B and C in hexaploid _E. crus-galli_; DH, EH, and FH, subgenomes D, E and F in hexaploid _E. colona_; aA and aB, ancestral diploid genomes of subgenomes A and B. Subgenomes from the same species are linked by color-coded lines. **e** The gene-tree quartet frequency around the divergence node a in the _Echinochloa_ phylogeny. Red (t1) represents the ASTRAL and concatenation species topology; blue (t2) and green (t3) trees represent two alternative topologies. The colored bars on the right panel show the frequencies of three topologies. Source data are provided as a Source data file.

previously[10]. Here, we further generated and used Hi-C data (122.8 Gb for _E. oryzicola_ with ~117-fold genome coverage and 181.0 Gb for _E. crus-galli_ with ~129-fold coverage) to anchor the contigs into chromosomes by using DipHiC (Supplementary Figs. 5 and 6; Supplementary Note 3 and 4). In total, 97.9% and 96.2% of sequences were anchored onto 18 and 27 chromosomes of _E. oryzicola_ and _E. crus-galli_, respectively (Supplementary Figs. 7 and 8). The subgenome-specific 13-mer and transposon element analysis further supported the accuracy of subgenome distinguishing for the two polyploid genomes (Supplementary Fig. 9). In total, 502.3 and 416.7 Mb contigs were assigned to the

two _E. oryzicola_ subgenomes AT and BT (where 'T' refers to tetraploid), and three _E. crus-galli_ subgenomes of sizes 473.2 (AH), 399.0 (BH), and 416.8 Mb (CH) were determined (Table 1). Centromeric regions were also found in all chromosomes (Supplementary Fig. 8b).

Among the nine subgenomes of the four _Echinochloa_ species, including the three polyploids used in this study and the diploid _E. haploclada_ genome (hereafter 'Ehap'), the subgenome sizes varied from 311 Mb (EH) to 502 Mb (AT), mostly resulting from the alterations of repeat elements (Table 1), which accounted for 36.6% (EH) to 53.7% (AH) of the subgenome size. The majority

**Table 1 Summary of the assembly and annotation of *Echinochloa* reference genomes.**

| Species | Karyotype | (Sub)genome | Assembly size (Mbp) | Number of annotated genes | TEs (%) |
|---|---|---|---|---|---|
| *E. oryzicola* | 2n = 4× = 36 | Whole genome | 945.6 | 68,544 | 51.5 |
| | | AT | 502.3 | 33,635 | 53.5 |
| | | BT | 416.7 | 33,316 | 46.4 |
| *E. crus-galli* | 2n = 6× = 54 | Whole genome | 1339.8 | 104,780 | 47.7 |
| | | AH | 473.2 | 33,261 | 53.7 |
| | | BH | 399.0 | 34,124 | 45.2 |
| | | CH | 416.8 | 35,808 | 45.0 |
| *E. colona* | 2n = 6× = 54 | Whole genome | 1127.2 | 93,697 | 44.5 |
| | | DH | 436.4 | 33,538 | 50.4 |
| | | EH | 310.5 | 30,593 | 36.6 |
| | | FH | 347.1 | 29,510 | 43.4 |

of repeat elements were *Gypsy*-type long terminal repeat (LTR) elements (Supplementary Fig. 10).

A total of 68,544, 104,780, and 93,697 genes were predicted in the *E. oryzicola*, *E. crus-galli*, and *E. colona* genome assemblies, respectively (Table 1). Compared with the diploid *E. haploclada* (37,510 genes), the polyploid subgenomes showed gene losses. Compared to other Poaceae species (Supplementary Data 1), except for *Oropetium thomaeum* (L.f.) Trin., which has the smallest known genome size in this family at ~285 Mb[24]; *Echinochloa* is characterized by marked contraction in the sizes of gene families related to biotic stress response (e.g., NB-ARC domain, legume lectin, D-mannose-binding lectin, UDP-glucoronosyl and UDP-glucosyl transferase) (Fig. 1c; Supplementary Fig. 11). This is probably a result of fitness costs associated with maintaining these stress response[25]. Interestingly, across the Poaceae, the NB-ARC gene family showed a high coefficient of variation (CV) in size, followed by other disease-related gene families (Supplementary Fig. 12). The evolutionary dynamics of the NB-ARC gene family size has also been observed in other plants (e.g., *Phaseolus*)[26]. Additionally, the number of NB-ARC genes generally changed linearly along with the diploid genome size ($R^2 = 0.69$). This is in contrast to the dramatic copy number variations in polyploid subgenomes, where the NB-ARC family was expanded in *Triticum* and contracted in *Echinochloa*, although the two polyploids shared similar formation processes (Fig. 1c). The NB-ARC genes were significantly enriched on chromosome 4 of *Echinochloa* ($P < 1 \times 10^{-5}$, Fisher's exact test), which is syntenic to chromosome 11 of rice[27] (Supplementary Fig. 13). Chromosome 4 of *Echinochloa* and chromosome 11 of rice were also found to have the highest transposon density in both genomes. It has been speculated that the abundance of transposable elements may have contributed to the copy number dynamics of the NB-ARC gene families across the Poaceae, facilitating rapid adaptation to local environments[28]. In contrast, gene families related to abiotic stress response (e.g., the AP2, C2, GRAS, and MYB-like DNA-binding domain families) showed conservation in size, with relatively low CVs across the Poaceae (Supplementary Figs. 11 and 12). Nonetheless, despite apparent genomic contractions after polyploidization, several local expansions, possibly underlying specific adaptations, were observed in *Echinochloa*. For example, an expansion from BT to BH resulting from the massive local duplication of two genes, one encoding an O-methyltransferase that plays a key role in lignin biosynthesis, stress tolerance, and disease resistance in plants[29], and the other encoding a pyridoxal-dependent decarboxylase involved in plant nitrogen metabolism[30] (Supplementary Fig. 4b).

**Phylogeny of the *Echinochloa* genus.** We first calculated the synonymous substitutions per synonymous site ($K_s$) of syntenic gene pairs among the *Echinochloa* subgenomes and other genomes of evolutionarily neighboring species, including *O. sativa* L., *Sorghum bicolor* (L.) Moench, *Digitaria exilis* (Kippist) Stapf (separated into two subgenomes *D. exilis*-A and *D. exilis*-B), *S. italica* (L.) P. Beauv., and *Panicum hallii* Vasey (Supplementary Fig. 14a). The peaks of $K_s$ distributions are regarded as divergence events. The genus-level phylogeny, which was built based on the $K_s$ distribution, was consistent with previous phylogenetic analyses using plastomes across the entire Poaceae[31] (Supplementary Fig. 14b). Assuming an average substitution rate of $6.5 \times 10^{-9}$ substitutions per year per synonymous site[10], *Echinochloa* and *Digitaria* were estimated to diverge at approximately 20.0 million years ago (mya), and the divergence of *Echinochloa* from *Setaria* and *Panicum* occurred ~14.6 mya, in line with the previous estimation[10]. Within *Echinochloa*, the LTR insertion time estimation dated the tetraploidization event of *E. oryzicola* to be ~2.0 mya (Supplementary Fig. 15), and the hexaploidization time of *E. crus-galli* was ~0.3 mya according to the $K_s$ distributions between AH (BH) and its ancestral subgenome AT (BT) (Supplementary Fig. 14). Details of the hexaploidization of *E. colona* could not be uncovered because of the lack of parental genomes, but subgenome DH in *E. colona* showed a fairly close genetic distance to CH in *E. crus-galli*, with an estimated divergence time of ~0.3 mya, overlapping with the hexaploidization time of *E. crus-galli*. This suggests that the ancestral diploid of CH is genetically close to the ancestral genome of DH and, therefore, *E. colona* is a neo-hexaploid species, just like *E. crus-galli* (Fig. 1d; Supplementary Fig. 14). In *Echinochloa*, subgenome A (including AT and AH) diverged from the other subgenomes ~4.6 mya. However, the $K_s$ distribution peaks between subgenome FH and subgenome A (including AT and AH) and between subgenome FH and Ehap, overlapped with the $K_s$ peak between subgenome A and Ehap, implying radiative evolution at the first divergence in *Echinochloa* (Supplementary Fig. 14).

We reconstructed the maximum-likelihood (ML) phylogeny based on a concatenated matrix of 3557 single-copy orthologs among the 16 (sub)genomes and also a coalescent-based phylogeny by integrating these single-copy gene trees (Fig. 1d; Supplementary Fig. 16a, b). Consistent with the result of the $K_s$ distribution analysis, a weak support existed at the node where FH diverged from the other subgenomes in the coalescent tree, confirming radiative evolution. Using single-copy orthologs, we counted the numbers of three gene topologies among AH, FH, Ehap, and an outgroup (one species tree and two incongruent trees for a triplet with an outgroup). The 1:1:1 ratio in the gene counts of the three trees implied possible incomplete lineage sorting (ILS) at the first *Echinochloa* divergence (Fig. 1e; Supplementary Fig. 16c). Subsequently, quantifying introgression via branch lengths (QuIBL), a method based on the internal branch length, was used to infer the proportions of potential ILS

or introgression event contributing to topological incongruence[32]. The results revealed that the discordance between the gene and species trees in *Echinochloa* was more likely caused by ILS rather than introgression, similar to the *Oryza* genus[33] (Supplementary Data 2). In addition, the unbiased distribution of three topologies of triplet (AH, FH, and Ehap) further supported the possibility of ILS in *Echinochloa* (Supplementary Fig. 17). To trace the ILS footprints along with the evolutionary process, we measured the topological discordance in various triplets of *Echinochloa* subgenomes (Supplementary Fig. 16d). With an increase in divergence time interval, the proportion of topological discordance decreased at the rate of 10.6% per million years at the whole-genome level in *Echinochloa* (Supplementary Fig. 16d). Conclusively, the case in *Echinochloa* confirms the substantial impact of ILS on phylogenetic inference, especially for groups that experienced rapid radiative evolution.

We noticed that *Digitaria* and *Setaria-Panicum* together formed a monophyletic group (clade) in the concatenated- and coalescent-based phylogeny, rather than a polyphyletic relationship, as inferred from the $K_s$ distribution analysis (Supplementary Figs. 14c, 16a, b). Specifically, the $K_s$ peak of *Setaria-Echinochloa* or *Panicum-Echinochloa* shifted to the left of *Digitaria-Echinochloa* (nodes 4 and 3 in Fig. 1d; Supplementary Fig. 14). To validate whether introgression caused incongruence, triplet gene trees were built (Supplementary Fig. 18). In total, only 40.03–41.64% of the gene trees supported the species tree in the triplet of *Setaria*, *Digitaria*, and *Echinochloa* with *Oryza* as an outgroup, although the time interval between the *Digitaria* and *Setaria* speciation times was greater than 5 million years, indicating that genetic exchange events (i.e., introgression), rather than ILS, drove topological incongruence. Additionally, 66.19% of gene trees matched the species topology in the triplet (*Sorghum*, *Digitaria*, and *Echinochloa*) with *Oryza* as an outgroup, smaller than the proportion of ~79%, estimated based on the triplet divergence interval of 3.5 million years under the condition of only ILS (Supplementary Fig. 16d). Frequent introgression among the genera was confirmed by QuIBL analyses (Supplementary Data 2). Taken together, these analyses indicate that both introgression and ILS play important roles in generating topological discordance between gene and species trees in the Paniceae tribe.

**Population structure and demography of global *Echinochloa* species**. To explore the diversity and population structure of *Echinochloa* on a global scale, a total of 707 accessions of barnyard grasses from rice paddy ecosystem (paddy fields and surroundings), including 328 accessions that have been sequenced in our previous study[6] and 409 newly sequenced in this study, and 30 cultivated accessions of barnyard millets from 16 countries were collected and used in genomic investigation (Fig. 2a; Supplementary Data 3). Clean reads were first mapped to the *E. crus-galli* (STB08) reference genome. Based on the read mapping rates and genome coverage (Fig. 2b; Supplementary Fig. 19), together with the morphological traits and genome sizes estimated by flow cytometry or *k*-mer analysis (Supplementary Data 3), the 737 accessions were classified into four groups, i.e., hexaploid *E. crus-galli* and *E. colona*, and tetraploid *E. oryzicola* and *E. walteri*. On average, 97.3% reads of the *E. crus-galli* group, 95.4% of *E. oryzicola*, 84.9% of *E. walteri*, and 70.8% of *E. colona* were mapped to STB08 (Fig. 2b). For the subgenomes AH and BH, the average read coverages for *E. crus-galli* accessions were 93.5% and 93.2%, respectively, which were higher than those for the accessions of *E. oryzicola* (87.6% and 86.8%), *E. walteri* (73.9% and 71.4%), and *E. colona* (25.0% and 40.1%), consistent with the fact that *E. oryzicola* is an ancestral parent of *E. crus-galli*. For subgenome CH,

the average read coverage for *E. crus-galli* accessions was 95.6%, while it was only 11.9% and 12.6% for *E. oryzicola* and *E. walteri* accessions, respectively, because of their lack of subgenome CH. However, *E. colona* accessions had 78.0% average read coverage on subgenome CH, indicating that subgenome DH of *E. colona* is genetically close to subgenome CH of *E. crus-galli* (Fig. 1d).

Our global collection was composed of 596 accessions of hexaploid *E. crus-galli*, 32 of hexaploid *E. colona*, 85 of tetraploid *E. oryzicola*, 15 of tetraploid *E. walteri*, and nine outliers (with abnormal read mapping rates or genome coverage; see details in next section). The accessions from China (90.0%), Brazil (86.7%), and Malaysia (94.7%) were dominantly *E. crus-galli*, while those from Italy were mainly *E. oryzicola* (89.3%). *Echinochloa* accessions from the United States were diverse, with the highest Simpson's diversity index of 0.676, compared to 0.182 from China, 0.100 from Malaysia, 0.231 from Brazil, and 0.191 from Italy (Fig. 2a).

We called single nucleotide polymorphisms (SNPs) for subgenome A (9.55 million and 20.2 SNPs/kb), B (6.56 million and 16.4 SNPs/kb), and C (4.60 million and 11.0 SNPs/kb) and used them in population structure and phylogenetic analyses for identifying varieties in each group (Fig. 2c; Supplementary Fig. 20). In *E. oryzicola*, a clade from Hainan, China (designated *E. oryzicola* var. *hainanensis*), showed signs of genetic admixture, despite similar read mapping rates and genome coverage as the other ordinary *E. oryzicola* accessions (designated *E. oryzicola* var. *oryzicola*). The accessions of the *E. oryzicola* var. *hainanensis* clade showed distinctive morphological characteristics, including prostrate tillers, small awnless spikelets, and purple panicles, and thus were easily distinguishable from *E. oryzicola* var. *oryzicola* accessions, which showed compact plant architectures, large spikelets with long awns, and green panicles (Fig. 2d; Supplementary Figs. 21 and 22; Supplementary Note 1). In *E. crus-galli*, five varieties (four weedy forms, including var. *crus-galli*, var. *crus-pavonis* (Kunth) Hitchc., var. *praticola* Ohwi, and var. *oryzoides* (Ard.) Lindm., and one cultivated form, i.e., var. *esculenta*) and two admixture groups could be determined (Fig. 2c; Supplementary Fig. 20). In the weedy forms, var. *crus-galli* and var. *oryzoides* usually have smaller tiller angles as a typical feature of rice mimicry (Fig. 2d; Supplementary Fig. 22). Var. *oryzoides* has the largest grain size or weight among all *E. crus-galli* varieties, even comparable to that of *E. oryzicola* var. *oryzicola* (Fig. 2d; Supplementary Fig. 21). The population structure clustering was supported by a principal component analysis (Supplementary Fig. 23). Although *E. crus-galli* was dominant in China, Brazil, and Malaysia, at the variety-level, var. *crus-galli* was predominant in paddy fields of China, while var. *crus-pavonis* was prevalent in Brazil and admix1 in Malaysia. In China, the northwestern and southern paddy fields were mainly infested by var. *oryzoides* and var. *crus-galli*, respectively (Supplementary Data 3).

The demographic history of *E. crus-galli* was further inferred using fastsimcoal2 based on the unfolded joint site frequency spectrum (SFS) of unlinked SNPs synonymous with known ancestral alleles[34]. Among the five divergence history scenarios, the best-fit scenario was the hybrid origin of *E. crus-galli* var. *crus-pavonis* with continuous gene flow from var. *crus-galli* to var. *crus-pavonis* (Fig. 2e; Model 5 in Supplementary Fig. 24 and Supplementary Table 1) according to the Akaike information criterion (AIC) score. The speciation of *E. crus-galli* was inferred to have occurred 283 thousand years ago, consistent with the estimation based on $K_s$. Var. *crus-pavonis* originated from the admixture between var. *crus-galli* (20.2%) and common ancestors (79.8%) of var. *praticola*, var. *oryzoides* and var. *esculenta* 183 thousand years ago, corresponding to the admixed genetic components in population structure when $K = 3$ and 4

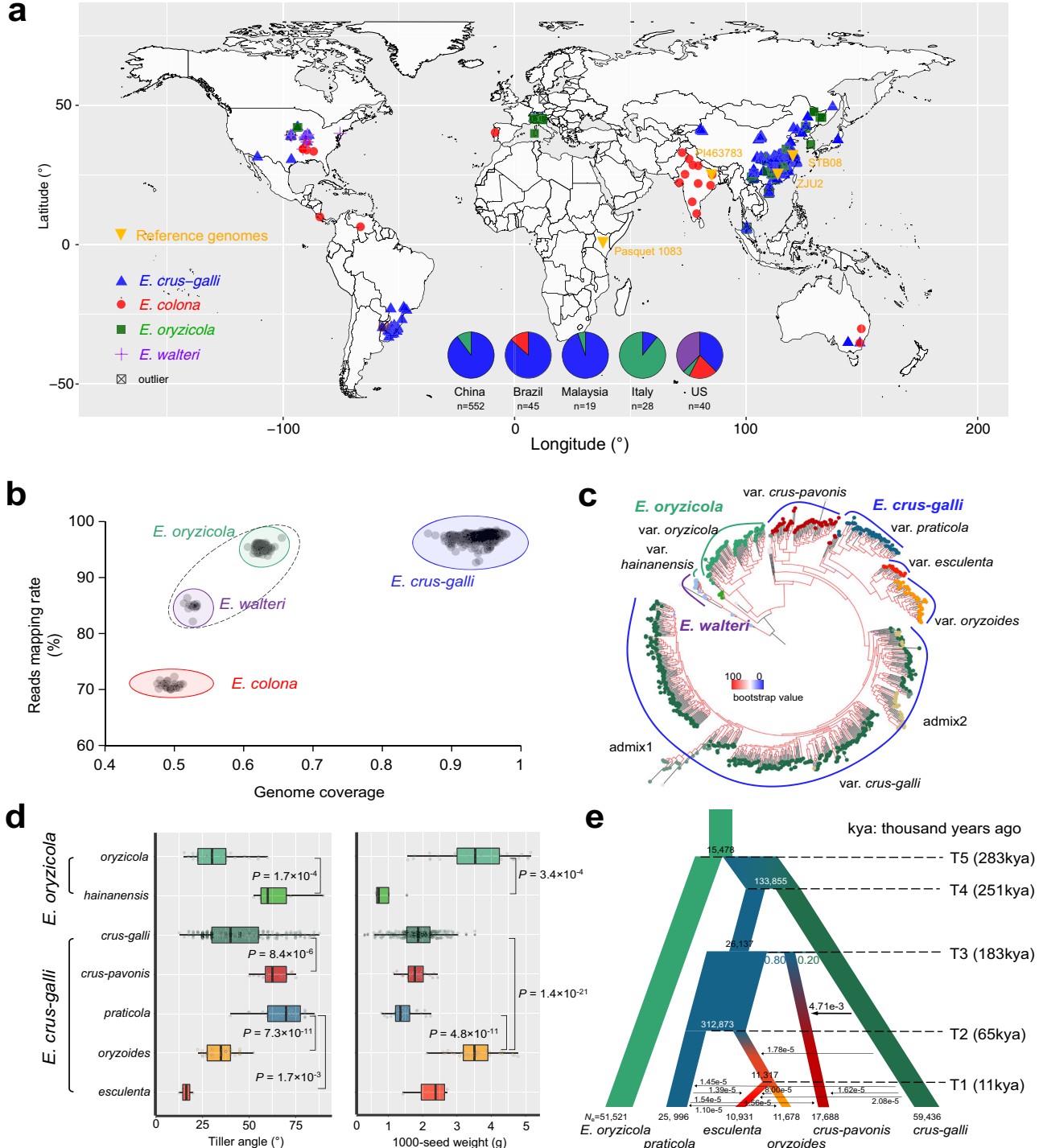

**Fig. 2 Population structure and demography of *Echinochloa* species. a** Global distribution of the *Echinochloa* plants used in this study. The locations of the four *Echinochloa* accessions with sequenced reference genomes are indicated by yellow downward triangles. The species distributions of the *Echinochloa* accessions sampled from five countries (China, Brazil, Malaysia, Italy, and United States) are shown in pie charts. **b** Four *Echinochloa* species are identified based on mapping rates and genome coverage of reads mapped to the *E. crus-galli* reference assembly (STB08). **c** The maximum-likelihood phylogeny of *E. crus-galli* and *E. oryzicola* accessions are inferred using genome-wide non-synonymous single nucleotide polymorphisms (SNPs), with tetraploid *E. walteri* as an outgroup species. **d** Phenotypic differences (tiller angle and 1000-seed weight) in the varieties of *E. oryzicola* and *E. crus-galli* (accession number: $n = 38$ *oryzicola*, $n = 6$ *hainanensis*, $n = 321$ *crus-galli*, $n = 15$ *crus-pavonis*, $n = 25$ *praticola*, $n = 38$ *oryzoides*, $n = 4$ *esculenta*). In the box plots, the horizontal line shows the median value, and the whiskers show the 25% and 75% quartile values of tiller angles and seed weights. *P* values are calculated using the two-tailed Wilcoxon test. **e** Demographic history of *E. crus-galli* varieties. T1 to T5 indicate the estimated time of population divergence. $N_e$, effective population size. Kya, thousand years ago. Source data are provided as a Source data file.

(Supplementary Fig. 20). The ancestral population of var. *oryzoides* and var. *esculenta* diverged from var. *praticola* 65 thousand years ago, and var. *esculenta* was domesticated 11 thousand years ago. In addition, frequent gene flow events were detected among varieties, especially after the emergence of agriculture (~10 thousand years ago) (Fig. 2e; Supplementary Fig. 25; Supplementary Data 4). Considering that barnyard grasses are crop companions, especially of rice, these results suggest that the agricultural activities and germplasm exchange via trading may have facilitated their dispersal and gene flow among varieties[35].

Recent history within the last 10 thousand years was examined based on both folded and unfolded SFS using Stairway Plot[36] (Supplementary Fig. 26). A sharp bottleneck was observed in var. *praticola* at ~3000 years ago. Population sizes were constrained, beginning at 1000 years ago for var. *crus-pavonis* and 400 years ago for var. *crus-galli*. After recovering from a bottleneck around 10,000 years ago, the population size of var. *oryzoides* remained stable in the recent 2000 years. Within the last 100 years, the effective population size of cultivated var. *esculenta* decreased severely and continuously.

Generally, an asymmetric distribution of nucleotide diversity ($\pi_A > \pi_B > \pi_C$) was observed in *Echinochloa* species (Supplementary Fig. 27a, b), similar to that in wheat[37]. For example, subgenome C showed the lowest diversity among the three subgenomes in *E. crus-galli* varieties, similar to the lowest diversity observed in subgenome D of wheat. This could be explained by the severe bottleneck in subgenome C during hexaploidization or increased diversity in subgenomes A and B after the speciation of *E. crus-galli*. In contrast, the differences between the diversities of subgenome A or B and subgenome C were larger in var. *crus-galli* than in the other four varieties, implying that var. *crus-galli* experienced more dynamic genomic alterations, which increased the nucleotide diversity of subgenomes A and B (Supplementary Fig. 27).

**Sympatric gene flow among *Echinochloa* weeds.** Interspecific introgression from ancestral populations is known to have played a primary role in increasing genetic diversity in various genera. For example, the genetic diversity of hexaploid bread wheat was enhanced by introgression from tetraploid emmer and other wheats[37,38]. We constructed an ML phylogenetic tree using SNPs in *Echinochloa* chloroplast genomes with *S. italica* as an outgroup (Fig. 3a). *Echinochloa colona* formed a monophyletic group (clade 0) and was located at the root. The remaining *Echinochloa* taxa were clustered into two clades (clade 1 or clade 2). *Echinochloa haploclada*, *E. walteri*, *E. oryzicola* var. *hainanensis*, and most *E. crus-galli* accessions were clustered in clade 2, while *E. oryzicola* var. *oryzicola* and some *E. crus-galli* var. *crus-galli* accessions were clustered in clade 1, although var. *crus-galli* in clade 1 and clade 2 was mixed together in the nuclear tree (Figs. 3a and 2c). It was previously reported that *E. oryzicola* var. *oryzicola* was assumed to be a male donor in the polyploidization of hexaploid *E. crus-galli*[16]. Here, the chloroplast phylogeny indicated that at least two male donors contributed to the origin of *E. crus-galli*. This is discordant with the hypothesis of a single origin inferred from nuclear genome phylogeny. Chloroplast capture may underlie the polyphyletic plastid phylogeny. Either sexual hybridization or asexual grafting can lead to chloroplast capture or horizontal transfer[39]. While natural grafting is common in woody plants, it has never been reported in grasses[40], whereas interspecific hybridization has been frequently reported in grasses.

To examine for potential introgression (gene flow) from *E. oryzicola* var. *oryzicola* to *E. crus-galli* var. *crus-galli*, four topologies were set in ABBA-BABA statistics (Fig. 3b; Supplementary Fig. 28).

Compared to the null hypothesis (T1), significantly higher $f_d$ values (the proportion of introgression) were observed across the nuclear genome in topologies T2, T3, and T4 (Supplementary Fig. 28). The proportions of genome introgression estimated by PGI (an index to reflect genome-wide introgression level)[37] were higher in topologies T2, T3, and T4 compared to the null topology. The relationship between windowed $f_d$ and nucleotide diversity $\pi$ confirmed that introgression has elevated the overall genetic diversity of sub-genomes A and B in var. *crus-galli*, with $\pi_A = 1.61 \times 10^{-3}$ and $\pi_B = 1.36 \times 10^{-3}$ when $f_d > 0.02$, compared to $\pi_A = 1.14 \times 10^{-3}$ and $\pi_B = 1.07 \times 10^{-3}$ when $f_d < 0.02$. However, a proportion of windows with $f_d$ values close to 1 showed extremely low $\pi$ values, suggesting that the fixation of introgressed segments would not increase diversity; instead, the median magnitude of introgression would increase to the maximum extent, similar to the pattern in wheat (Supplementary Fig. 29).

Genes related to stress response were enriched in introgressed segments with high frequency ($f_d > 0.5$) (Supplementary Fig. 30). For example, a large region from 7.0 Mb to 9.0 Mb on chromosome AH05 had an extreme $f_d$ value close to 1. This region has a total of 34 genes, including seven biotic and 12 abiotic stress response genes (Fig. 3b).

*Echinochloa oryzicola* var. *oryzicola* and *E. crus-galli* var. *crus-galli* have overlapping distributions and ecological niches (Supplementary Data 3; Supplementary Note 1). Three *E. crus-galli* accessions (QX6, JX9, and JX111) from China were scattered within *E. oryzicola* var. *oryzicola* in the chloroplast tree, reflecting the recurrent hybridization events between *E. oryzicola* var. *oryzicola* and var. *crus-galli* (Fig. 3a). To examine whether they were recent hybrids, re-sequenced reads were mapped to a decaploid pseudo-genome created by combing the STB08 and ZJU2 reference assemblies. For *E. oryzicola* var. *oryzicola* and *E. crus-galli* var. *crus-galli* accessions without introgression, the mapping depths in the five subgenomes were biased (Fig. 3c), while for accessions JX111 and QX6, the subgenome mapping depths were symmetric, clearly demonstrating the recent hybridization between *E. oryzicola* and *E. crus-galli*. However, for JX9, the biased mapping depths revealed that small introgression footprints were retained in nuclear genomes as a result of back-crossing with *E. crus-galli*, after an ancient interspecies hybridization event, which led to chloroplast genome substitution. These results were consistent with admixed genomic compositions revealed by population structure analysis (Supplementary Fig. 31). Based on this method, we found that six of the nine outliers (identified by the abnormal rate of read mapping and genome coverage against *E. crus-galli* reference genome STB08) in our collection were recent hybrids with multiple subgenomes from more than one species (Supplementary Data 3; Supplementary Fig. 31).

**Local adaptation to natural environments.** Distribution of *E. crus-galli* varieties showed high differentiation with respect to habitat, with var. *crus-galli* and var. *crus-pavonis* mainly found in low altitudes, while var. *praticola* and var. *oryzoides* typically occurring in relatively high altitudes (Supplementary Fig. 32). A difference in latitudinal distribution was also observed in *E. oryzicola*, with accessions from higher latitudes (Italy, Northeast China, Korea, and Japan) forming a clade (hereafter referred to as group HL) separated from those occurring in lower latitudes (group LL) (Southern China and Malaysia) (Supplementary Figs. 32 and 33). We calculated the genomic differentiation ($F_{st}$) between var. *crus-galli* and var. *praticola* in *E. crus-galli* and between group LL and HL in *E. oryzicola* to trace the footprints of local adaptation to natural environments and to explore the potential convergent evolution of the two *Echinochloa* species.

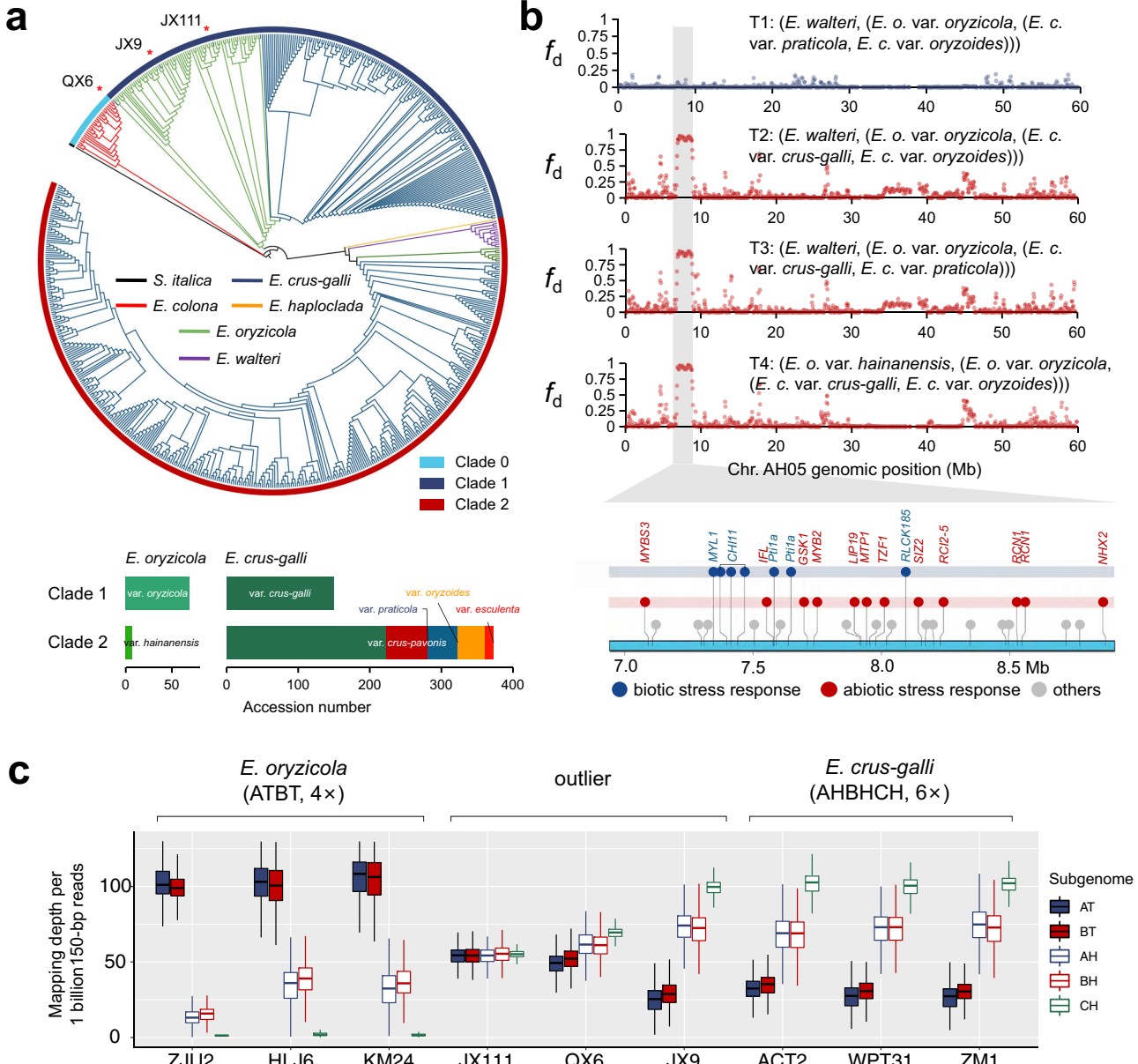

**Fig. 3 Sympatric gene flow among *Echinochloa* species. a** A maximum-likelihood phylogeny based on single nucleotide polymorphisms (SNPs) in the chloroplast genomes of *Echinochloa* accessions (*n* = 607). Three non-*E. oryzicola* accessions (QX6, JX111, and JX9) clustered within *E. oryzicola* var. oryzicola (clade 1) are indicated by red asterisks. Histogram charts below the phylogeny tree show the species and variety compositions in clade 1 and clade 2. **b** Proportion of introgression $f_d$ distributions of ABBA-BABA test on chromosome AH05 in topology T1–T4 (see details in Supplementary Fig. 28). Stress-responsive genes with sequences ranging from 7 Mb to 9 Mb on chromosome AH05 are shown in the lower panel. **c** Mapping depths when reads from three *E. oryzicola* accessions, three *E. crus-galli* accessions, and three outliers are mapped against the pseudo-decaploid reference genome (STB08 + ZJU2). In the box plots, the horizontal line shows the median value, and the whiskers show the 25% and 75% quartile values of mapping depths of 1-Mb sliding windows in five subgenomes (window number: *n* = 506 AT, *n* = 421 BT, *n* = 478 AH, *n* = 405 BH, *n* = 421 CH).

Flowering time (FT) is a crucial fitness trait in local adaptation. In *E. crus-galli*, var. *crus-galli* flowered significantly later than var. *praticola* and var. *oryzoides* ($P = 5.7 \times 10^{-11}$ and $P = 2.1 \times 10^{-161}$, respectively, Student's *t*-test), and in *E. oryzicola*, the accessions from group HL flowered earlier than those from group LL ($P = 2.6 \times 10^{-6}$, Student's *t*-test), as found in our common-garden experiments (Supplementary Fig. 32). A total of 13 genes (7 in subgenome A, 5 in B, and 1 in C) regulating FT were found to be highly differentiated (top 1% in $F_{st}$ windowed values) in *E. crus-galli*, including *FT*, *Hd1*, *Ehd1*, and *DTH2* (Fig. 4a; Supplementary Fig. 34; Supplementary Data 5). Among these genes, *Hd1*

(AH06.1150), located on chromosome 6 of subgenome A, was also identified during *E. oryzicola* differentiation (Fig. 4a; Supplementary Data 6). A test of selection (Tajima's *D*) and nucleotide diversity ($\pi$) analyses showed that a 9-Mb genomic region extending from 13 to 22 Mb on chromosome AH06, harboring *Hd1*, was under strong positive natural selection in var. *crus-galli* (Fig. 4b; Supplementary Fig. 35). A genome-wide association study (GWAS) for the FT of *E. crus-galli* also identified *Hd1* (Fig. 4b; Supplementary Data 7). Selection on *Hd1* was similarly detected in var. *crus-pavonis* and group LL that shared the late-flowering trait with var. *crus-galli*, implying that

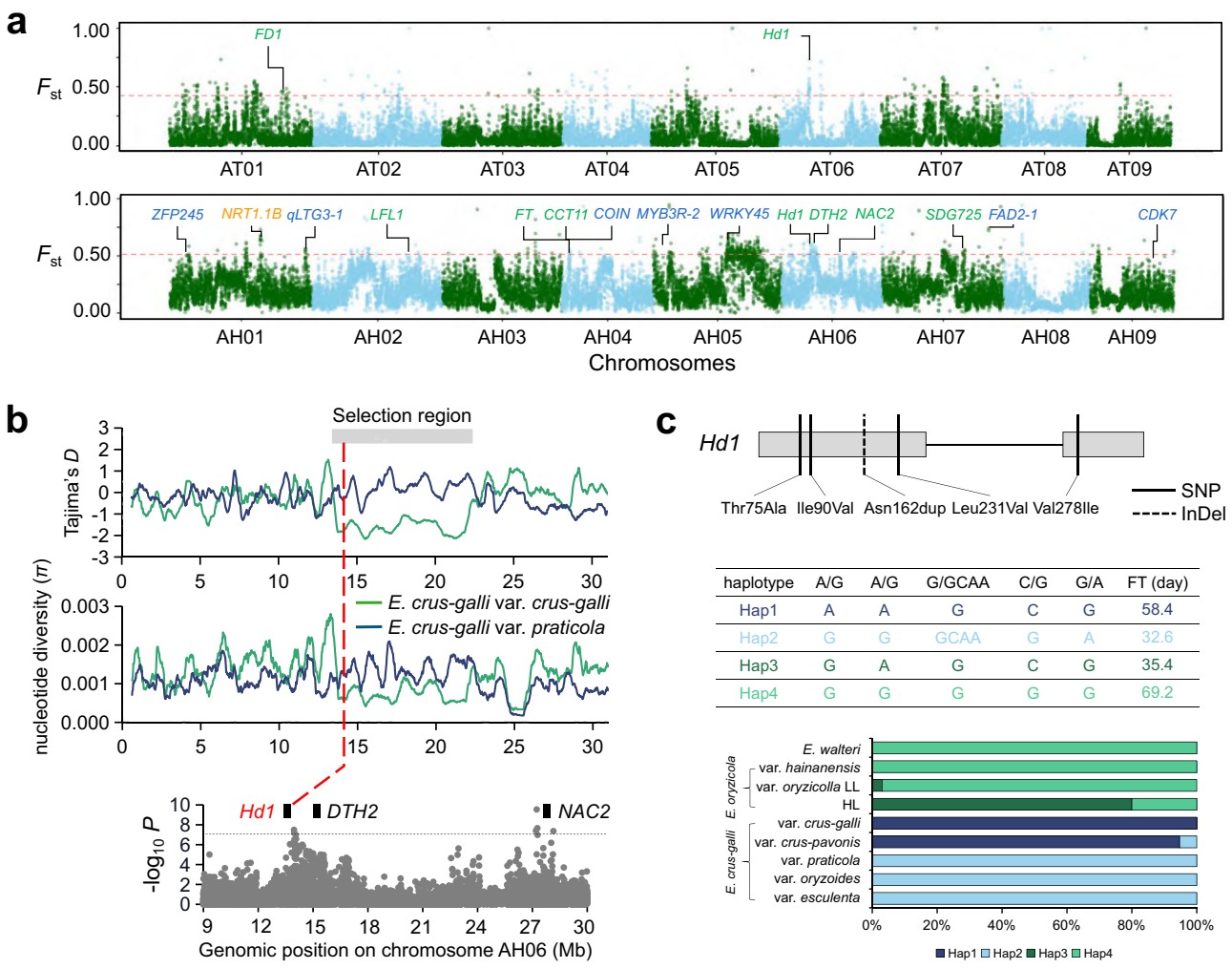

**Fig. 4 Local adaptation of *Echinochloa* accessions. a** Genomic differentiation ($F_{st}$) between high-latitude (HL) and low-latitude (LL) *E. oryzicola* var. *oryzicola* populations (upper panel), and between *E. crus-galli* var. *praticola* and var. *crus-galli* (lower panel). Genes related to flowering (green) and cold tolerance (blue) in the top 1% $F_{st}$ windows (red horizontal dashed lines) in *E. oryzicola* and *E. crus-galli* subgenomes A are shown, respectively. **b** Selection features (Tajima's *D* and nucleotide diversity $\pi$) of *Hd1* and genome-wide association study on flowering time in *E. crus-galli*. The red dashed line indicates the genomic position of *Hd1* on chromosome AH06. **c** Analysis of *Hd1* haplotypes in *E. oryzicola* and *E. crus-galli*. Four haplotypes composed by four SNPs and one InDel show differences in flowering time. Source data are provided as a Source data file.

*Hd1* plays essential roles in regulating FT in *Echinochloa*. Given that *E. crus-galli* and *E. oryzicola* are polyploids, the profiling of *Hd1* genes in other subgenomes was scanned. No obvious signal of differentiation (indicated by $F_{st}$) and selection (indicated by Tajima's *D* and $\pi$) was found for the *Hd1* genes (BH06.1184 and CH06.1302) in subgenomes B and C, suggesting preferential evolution among subgenomes of *Echinochloa* in environmental adaptation (Supplementary Fig. 36).

We identified four *Hd1* haplotypes, composed of five functionally important and divergent non-synonymous polymorphisms (four SNPs leading to amino acid substitution and one 3-bp insertion leading to an Asn duplication) (Fig. 4c). Accessions with Hap1 and Hap4 showed significantly later flowering (58.4 and 69.2 days, respectively) relative to those with Hap2 and Hap3 (32.6 and 35.4 days, respectively) ($P = 3.6 \times 10^{-32}$ between Hap1 and Hap2, $P = 4.1 \times 10^{-9}$ between Hap3 and Hap4, Student's *t*-test) (Fig. 4c). Hap3 and Hap4 were only found in tetraploid *E. walteri* and *E. oryzicola*, and Hap1 and Hap2 only in *E. crus-galli*. These observations imply that *Hd1* functioned in parallel in the natural adaptation of *E. oryzicola* and *E. crus-galli*.

Furthermore, it is worth noting that var. *praticola* is predominantly found in dry lands, compared to var. *crus-galli*, var. *oryzicola*, and var. *crus-pavonis*, which occur in the lowlands, especially paddy fields (Supplementary Note 1). Besides differentiation at the level of macroscopic climate adaptation between var. *praticola* and var. *crus-galli*, genes underlying habitat preference may also be divergent. Gene ontology (GO) terms related to stress response (GO:0006979, response to oxidative stress, $P = 0.042$; GO:0009415, response to water, $P = 0.0049$) and nitrogen utilization (GO:0008519, ammonium trans-membrane transporter activity, $P = 0.022$; GO:0015696, ammonium transport, $P = 0.022$) were enriched in highly differentiated genomic regions. A homolog of *NRT1.1B* (AH01.2967) located on chromosome AH01 showed high differentiation potential ($F_{st} = 0.63$) (Fig. 4a). *NRT1.1B* has been reported as a key transporter in regulating differential nitrogen utilization between *japonica* and *indica* rice types[41]. The haplotype of *NRT1.1B* diverged between var. *praticola* and other varieties of *E. crus-galli*, with four amino acid substitutions, suggesting that var. *praticola* may have a differential nitrogen utilization ability compared to other varieties, allowing it to grow well in low-nitrogen-content uplands (Supplementary Fig. 37).

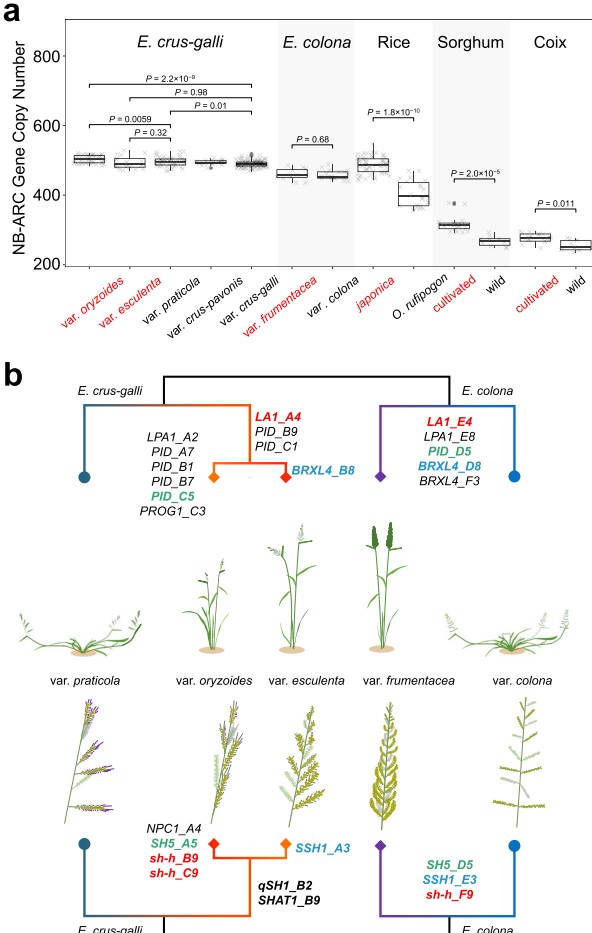

**Fig. 5 Domestication of barnyard millets. a** Alteration of the NB-ARC gene family size during domestication for five crops (two barnyard millets, rice, sorghum and coix). *E. crus-galli*: $n = 43$ *oryzoides*, $n = 18$ *esculenta*, $n = 48$ *praticola*, $n = 22$ *crus-pavonis*, $n = 115$ *crus-galli*; *E. colona*: $n = 11$ *frumentacea*, $n = 19$ *colona*; rice: $n = 55$ *japonica*, $n = 22$ wild; sorghum: $n = 18$ cultivated, $n = 10$ wild; Coix: $n = 18$ cultivated, $n = 9$ wild. In the box plots, the horizontal line shows the median value, and the whiskers show the 25% and 75% quartile values of NB-ARC gene copy numbers in five species. *P* values are calculated using the two-tailed Wilcoxon test. **b** Genomic divergence of key genes regulating plant architecture and grain shattering between barnyard millets and grasses. Chromosome IDs are indicated after the gene names. For example, '*LA1_A4*' represents the gene *LA1* on chromosome AH04. Source data are provided as a Source data file.

**Facilitation of herbicide resistance gene discovery in barnyard grass populations**. ALS inhibitor herbicides are among the most widely used worldwide, and consequently resistance evolution (both target-site- and non-target-site-based) to these herbicides exceeds that of any other site-of-action herbicide[11,13,42,43]. Genome resequencing of the barnyard grass samples from the paddy fields of Brazil, Italy, and China revealed three known ALS resistance mutations (Ala122Thr, Trp574Leu, and Ser653Asn) and a Gly654Cys substitution that has not been reported previously in plants (Supplementary Data 8). Each of these ALS resistance mutations was associated with the resistance phenotype and occurred preferably in subgenome A (coincided with enriched TE) (Supplementary Data 8 and 9). In addition, most ALS resistance mutations were found to occur in a homozygous state (especially for the mutations at amino acid positions 122, 653, and 654); this apparent selection for homozygosity could

potentially be a mechanism to overcome dilution effects by the wild-type (susceptibility) homeo-alleles across the genome.

Quinclorac is a highly selective auxin herbicide for barnyard grass control in rice fields. Resistance to quinclorac in barnyard grasses renders this herbicide ineffective in rice-producing countries, including China, Brazil, and USA. Despite much effort, the molecular basis of quinclorac resistance remains elusive[44]. Recently, a point mutation in the conserved degron region of auxin receptor AUX/IAA16 and a 27-bp deletion in the degron tail region of AUX/IAA2 have been demonstrated to endow resistance to other auxin herbicides (e.g., 2,4-D) in weedy plant species[45,46]. In light of these discoveries and taking advantage of our genome data, a total of 60 AUX/IAA genes were identified in the *E. crus-galli* genome (domain prediction as Aux/IAA gene family with the degron region 'VGWPPV') and compared for SNPs and InDels (small insertions and deletions) between the Brazilian samples of resistant and susceptible phenotypes. An Arg86Gln mutation in the conserved degron tail region of *Echinochloa* AUX/IAA12 was identified in one sample (SA88) with the resistance phenotype (Supplementary Figs. 38 and 39), and it is a potential candidate target-site gene for quinclorac resistance, pending for further confirmation at the population, genetic, and functional levels.

Apart from target-site resistance, non-target-site resistance (NTSR) also endows herbicide resistance[47,48]. Indeed, some Brazilian and Italian samples with the multiple resistance phenotypes had no ALS and ACCase mutations, indicating NTSR to herbicides, usually relating to reduced herbicide uptake/translocation and enhanced herbicide metabolism, involving super families of ABC transporters, cytochrome P450 (CYP450), and glutathione S-transferase (GST)[47,48]. Convergent evolution of gene copy number variation (CNV) has been shown in target-site resistance to glyphosate in many weedy species[48,49]. To investigate whether NTSR to ALS inhibitors is related to CNV, we estimated the copy numbers of CYP450, GST, and ABC transporter genes in the Brazilian *E. crus-galli* samples lacking ALS target-site mutations and observed no amplification of CYP450 and GST genes but slight expansion of ABC transporter genes ($P < 0.05$, Student's *t*-test) (Supplementary Data 9), indicating the existence of other NTSR mechanisms that are not related to CNV. A glyphosate-detoxifying aldo-keto reducatse (AKR) and glyphosate-exporting ABCC8 transporter have been cloned and characterized from barnyard grass in our previous studies, and a higher expression of these genes was found to confer glyphosate resistance[50,51]. It is interesting to determine whether the CNV of these genes is involved and the underlying genomic mechanism. Our current study annotated 867 CYP450s, 227 GSTs, 113 AKRs, and 361 ABC transporters in the *E. crus-galli* genome, and 694, 226, 111, and 336 of each category in the *E. colona* genome. These are rich resources vital to gene discovery in weedy plants for stress adaptation in general and for herbicide NTSR in particular.

**Domestication of barnyard millets**. As noted above, domestication events occurred in at least two hexaploid *Echinochloa* species, *E. crus-galli* and *E. colona*. From the phylogenetic analysis results (Fig. 2c; Supplementary Fig. 40), we inferred that *E. crus-galli* var. *esculenta* and *E. colona* var. *frumentacea*, mainly cultivated in East Asia and India, respectively[19,52], were domesticated from *E. crus-galli* var. *praticola* and *E. colona* var. *colona*, respectively. Interestingly, despite being a paddy weed, *E. crus-galli* var. *oryzoides* has domestication-like traits, for instance, the largest grain size among all *E. crus-galli* varieties and wheat-like plant architecture (Fig. 5b; Supplementary Note 1).

Like other crops, nucleotide diversity in barnyard millets decreased during domestication. For example, in *E. crus-galli*, the

nucleotide diversities of var. *esculenta* ($\pi = 0.865 \times 10^{-3}$) and var. *oryzoides* ($\pi = 0.773 \times 10^{-3}$) were significantly lower than the nucleotide diversity of var. *praticola* ($\pi = 1.021 \times 10^{-3}$) (Supplementary Fig. 27). The genetic distances estimated for linkage disequilibrium (LD) decay (indicated by $r^2$) to half of the highest value were obviously longer across whole genomes in the two *E. crus-galli* domesticates (*E. crus-galli* var. *esculenta*, $r^2 = 0.32$, 28 kb; var. *oryzoides*, $r^2 = 0.29$, 35 kb) than in the weed populations (*E. crus-galli* var. *praticola*, $r^2 = 0.26$, 0.6 kb). However, the LD magnitudes were similar between the cultivated and wild/weed forms in *E. colona* (91 kb for *E. colona* var. *frumentacea* when LD decayed to $r^2 = 0.39$; 89 kb for *E. colona* var. *colona* when LD decayed to $r^2 = 0.37$) (Supplementary Fig. 41), implying a domestication impact of lower magnitude on the genome of *E. colona*.

Artificial selection during domestication and improvement has increased the diversity and size of the NB-ARC family to improve immune responses in rice[27], wheat[10], and foxtail millet (336 in foxtail grass *versus* 368 in foxtail millet) (Fig. 1d). We developed a method to infer the domestication magnitude of barnyard millets by comparing the NB-ARC family size in the accessions of cultivated and wild species based on the abundance of reads generated by high-depth resequencing (Supplementary Note 6; Supplementary Data 10). We first tested the method in rice, sorghum, and the orphan crop Job's tears (*Coix lacryma-jobi*) and achieved a result consistent with that generated based on comparisons using reference genomes in all three cases, i.e., the NB-ARC family size was significantly larger in cultivated species than in their corresponding wild progenitors (Fig. 5a). In these cases, the size of the NB-ARC family appears to increase with the level of domestication (i.e., *japonica* rice > sorghum > Job's tears). However, in barnyard millets, no increase was observed in the NB-ARC family size in either the cultivated *E. colona* var. *frumentacea* or the cultivated *E. crus-galli* var. *esculenta*, implying that their domestication intensity was low (Fig. 5a; $P = 0.011$ for coix, $P = 2.0 \times 10^{-5}$ for sorghum, and $P = 1.8 \times 10^{-10}$ for rice, Wilcoxon test). Detection of significantly greater number of NB-ARC genes in *E. crus-galli* var. *oryzoides* than in other *Echinochloa* varieties supports the speculation of *E. crus-galli* var. *oryzoides* being an ancient crop ($P = 0.0059$, Wilcoxon test).

We scanned the allele frequency spectrum of non-synonymous variation in crucial genes underling key domestication traits (e.g., plant architecture, shattering loss, and grain size)[53]. In the gravity response pathway associated with regulation of tiller angles, eight non-synonymous variations in four homologs (*LA1*, *BRXL4*, and two *PID* genes) were found to be highly divergent between *E. crus-galli* var. *esculenta* and *E. crus-galli* var. *praticola*, and 16 non-synonymous substitutions in nine homologs (*LA1*, *LPA1*, *PROG1*, and six *PID* genes) were found to be highly divergent between *E. crus-galli* var. *oryzoides* and *E. crus-galli* var. *praticola* (Fig. 5b; Supplementary Data 11). In *E. colona*, seven non-synonymous variations in five homologs (*LA1*, *LPA1*, *PID*, and two *BRXL4* genes) were divergent between the cultivated and weedy *E. colona*. We noticed that *LA1* on chromosome 4 was highly divergent in three domesticates, and the *LA1* homologs on chromosome AH04 and EH04 were in perfect syntenic relationship, suggesting that parallel selection targeted these genes during the domestication of *E. crus-galli* and *E. colona* (Supplementary Fig. 42). *LA1_A4* (*LA1* on chromosome AH04) was divergent from *E. crus-galli* var. *praticola* in the common ancestor of var. *esculenta* and var. *oryzoides* by four amino acid replacements (Lys107Glu, Leu211Phe, Arg290Lys, and Lys340Asn), while the divergence in *LA1_E4* of *E. colona* was due to another amino acid change (Arg229Cys) (Supplementary Data 11). The $F_{st}$ around the six *LA1* copies on chromosome 4 in the two species indicated that not only *LA1-A4*, but also *LA1_B4* and *LA1_C4* showed high

differentiation in the two *E. crus-galli* domesticates, in contrast to high differentiation on only chromosome EH04 in *E. colona* (Supplementary Fig. 42; Supplementary Data 12–14). Two syntenic *PID* copies on chromosome 5 were both selected in *E. crus-galli* var. *oryzoides* and *E. colona* var. *frumentacea*, and two syntenic *BRXL4* copies were shared by *E. crus-galli* var. *esculenta* and *E. colona* var. *frumentacea*. Both revealed parallel selection on syntenic homologs in independent domestication events (Fig. 5b; Supplementary Fig. 43; Supplementary Data 11). For shattering-related homologous genes, non-synonymous variants were found to have experienced selection in *sh-h*, *SH5*, *SSH1*, *SHAT1*, *qSH1*, and *NPC1* genes in three barnyard millets (Fig. 5b; Supplementary Fig. 43; Supplementary Data 11). In grain enlargement-related homologous genes, a total of 81 non-synonymous variants were found to be differentiated between barnyard millets and grasses in key genes controlling grain size, including *GS5*, *TGW6*, and *qGW8* (Supplementary Data 11).

## Discussion

Polyploidy plays an important role in plant genome evolution[54,55]. To date, the genomes of five hexaploids have been sequenced and bread wheat is the sole hexaploid member to have been assembled at the chromosome level[56,57]. The genomes of two allohexaploid *Echinochloa* species (*E. crus-galli* and *E. colona*) used in this study are among the most compact hexaploid genomes (<1.5 Gb) currently known in plants (https://cvalues.science.kew.org/), with less than one-tenth of the genome size of bread wheat (~17 Gb). They, therefore, are an ideal model system for studying plant polyploidization and the impact of the events, such as complex interactions among subgenomes and genomic dynamics following polyploid speciation. Despite the small genome size, generating an accurate chromosome-level assembly of allohexaploid *Echinochloa* is still a challenge, due to high level of sequence similarity between subgenomes and the presence of homeologous exchanges. Here we built genome assemblies of *Echinochloa*, although we believe that the assemblies could be further improved by integrating information from other approaches, such as genetic maps and optical maps.

Accurately distinguishing *Echinochloa* species is the first step for basic and applied research of the species, including precision weed control, and is still a challenge due to the morphological similarity among the diverse *Echinochloa* species[16]. We could accurately distinguish *Echinochloa* species and varieties by integrating morphological characteristics with multiple pieces of genomic evidence, including genome size, read mapping rate, genome coverage, phylogeny, and population structure (Fig. 2; Supplementary Note 1). As a result, we found not only diverse compositions of barnyard grasses in different rice-growing areas, but a new *Echinochloa* clade (designated *E. oryzicola* var. *hainanensis*) (Fig. 2d; Supplementary Note 1). Moreover, we speculate *E. crus-galli* var. *oryzoides* being an ancient crop that had once undergone domestication and was subsequently abandoned by farmers based on domestication-like morphological and genomic signatures, such as larger grain size, non-shattering spikelets, lower $\pi$ value, and longer LD decay, as compared to var. *praticola* (Supplementary Figs. 27 and 41; Supplementary Note 1). This is supported by the evidence that *E. macrocarpa*, a race of *E. crus-galli* var. *oryzoides*, has been reported to be harvested as a crop in the Caucasus[58].

Reticulate evolution occurs commonly in plant speciation, especially in polyploid species complexes[59]. Our whole-genome data analyses imply radiative evolution at the first divergence in *Echinochloa* and we considered it as an ILS event, based on short interval length and low bootstrap value from species phylogenies, balanced phylogeny proportion in triplet (AH, FH, and Ehap),

QuIBL test, and the unbiased distribution of three topologies of triplet (AH, FH, and Ehap) (Supplementary Figs. 14, 16, and 18; Supplementary Data 1). Demographic history of *Echinochloa* populations is consistent with farming practices (Supplementary Fig. 26). With ancient farming practices, weeds such as var. *praticola* (a sharp bottleneck at approximately 3000 years ago) with a prostrate plant architecture (a typical trait non-mimic to crops) were more likely to be distinguished and removed by hand weeding. Owing to the popularity of rice transplanting, instead of direct seeding, in China hundreds of years ago, a higher selection pressure was likely applied to barnyard grasses, plausibly resulting in var. *crus-galli* population contraction. A gap existed in seeding growth between rice and barnyard grass, although var. *crus-galli* showed a rice-mimic morphology at the seedling stage. The effective population size of cultivated var. *esculenta* decreased severely and continuously within the last 100 years, likely a result of the recent abandonment of barnyard millet cultivation thanks to improvement of rice planting technologies and management practices that ensured sufficient food supplies and resulted in the replacement of barnyard millets by cold-tolerant rice varieties[19,52].

Fundamental understanding of herbicide resistance-conferring genes and resistance evolution are important for the development of strategies to delay, mitigate, and manage herbicide resistance. Compared to diploids, resistance gene discovery is more challenging in polyploid weedy species, such as barnyard grasses, because of the presence of multiple homeo-alleles across subgenomes[60]. In this study, the genomic data of global barnyard grasses provide an opportunity to examine variations related to herbicide TSR and rich resources vital to gene discovery for NTSR. Due to exceptionally severe selective pressures, convergent adaptation across species and repeated evolution, herbicide resistance provides an outstanding opportunity to investigate key questions about the genetics of adaptation, in particular the relative importance of adaptation from new mutations, standing genetic variation, or gene flow, with an example of the well-researched herbicide resistance evolution in *Amaranthus tuberculatus*[61,62]. The abundant genomic resources of *Echinochloa* weeds produced in this study would accelerate the research on herbicide resistance evolution.

As typical orphan crops, barnyard millets show excellent resistance to abiotic stresses[7]. Genomic feature analyses revealed that barnyard millets were domesticated only to a low magnitude, in contrast to major crops, such as rice. This kind of 'light' domestication or a wild-/weed-like genome might be critical for orphan crops to adapt to extreme local environments. In the case of barnyard millets, they may have competitive advantages under unfavorable conditions if they have similar genomic features with barnyard grasses. The recently proposed de novo domestication concept of maintaining genetic diversity and environmental adaptability while improving yield potential of wild species, is in line with the genomic feature of orphan crops[63]. Another reasonable explanation is that orphan crops may be non-fully domesticated.

In summary, the genomes of the *Echinochloa* polyploids produced in this study provide reference genomes for barnyard grasses and millets. These reference genomes together with the profiling of global *Echinochloa* accessions detailed here will accelerate research in weed evolutionary biology, interaction between crop and weed, and development of novel weed control strategies under the future climate change scenarios.

## Methods

**Genome sequencing and de novo assembly.** Total DNA for genome sequencing was extracted from the young leaves of a cultivated *E. colona* accession PI463783 using the cetyltrimethylammonium bromide (CTAB) method. DNA libraries for single-molecule real-time (SMRT) PacBio genome sequencing were constructed following the standard protocols of the Pacific Biosciences Company and sequenced on the PacBio Sequel II platform using the circular consensus sequencing (CCS) approach. Raw reads were corrected using an error correction module. Hifiasm (v0.12-r304)[64] was used for the initial assembly using HiFi reads with parameters '-a 4 -n 3 -x 0.8 -y 0.2.' After mapping the long subreads to the initial assembly with minimap2 (v2.15-r905)[65], racon (v1.4.3)[66] was used for correction for three rounds under default parameters. Gmap (v2014-05-15)[67] was used to annotate the draft assembly using the coding sequences (CDS) of diploid *E. haploclada*, and the synteny between them was conducted according to the physical orders along chromosomes. The contig-level draft assemblies of *E. oryzicola* and *E. crus-galli* were generated by our previous study using PacBio continuous long-read (CLR) sequencing[10].

**Genome scaffolding.** Following the standard protocol, we constructed Hi-C libraries of *E. colona* using the fresh young leaves with the digestion of a 4-cutter restriction enzyme MboI. Hi-C sequencing libraries were sequenced on Illumina HiSeq 2500 platform with 2 × 150-bp reads. Valid ligation products were used in pseudo-molecule construction with DipHiC pipeline (see details at Supplementary Note 2). In the first module "subgenome distinguishing", we integrated three clustering algorithms to distinguish subgenomes. First, 100-mers split from diploid *E. haploclada* assembly were mapped to *E. colona* draft assembly, and the depth of each contig was calculated. According to the biased depths, contigs were clustered into cluster1, cluster2, cluster3, and un-clustered. Second, inter-contig synteny was investigated to construct the contig allelism, which was used to correct previous depth-clustering and re-cluster contigs. Third, by mapping pairs of Hi-C reads to draft assembly using Burrows-Wheeler Aligner (BWA; version 0.7.15-r1140)[68] and filtering, the linkages or interactions were computed among contigs and used for correcting the clustering in the previous step. In the second module 'chromosome building', directional interactions and pre-assignment of chromosome groups were utilized to cluster contigs on each chromosome. Finally, module *optimize* implemented in AllHiC[69] was used to order and orientate contigs within each chromosome.

The procedures of Hi-C library construction and HiC-assisted scaffolding for *E. oryzicola* genome were similar to *E. colona* (Supplementary Note 3). The pipeline of subgenome distinguishing in *E. crus-galli* was slightly different because of the availability of assemblies of two parental genomes (*E. haploclada* and *E. oryzicola*) (Supplementary Note 4). Besides, considering its assembly was based on CLR sequencing with a relative low contig N50, we first corrected the errors in contig assembly by integrating the genetic relationship with parental three subgenome and intra-contig interactions inferred from the Hi-C library and a 20-kb mate-pair library, to rule out the noise of contig mis-assembly. The genomes of *E. haploclada* and *E. oryzicola* were both used to distinguish subgenomes in *E. crus-galli*, and the contig order and orientation were the same as those used previously (Supplementary Note 4). The genome-wide Hi-C all-to-all interaction heatmaps were generated by AllHiC.

**Genome quality assessment.** To evaluate the completeness of *E. colona* assembly, BUSCO (v4.1.2)[70] and LTR assembly index (LAI) using LTR retrotransposons[71] were assessed. The Illumina short reads and RNA-Seq data were used to evaluate the assembly completeness and accuracy by Bowtie2 (v2.3.5.1)[72]. To evaluate the accuracy with which the subgenomes were distinguished, we identified subgenome-specific *K*-mers, which could be used to differentiate homoeologous chromosomes. *K*-mers (*K* = 13) were generated using jellyfish (v2.3.0) and the frequency of each *K*-mer was counted for each chromosome[73]. Only the *K*-mers that were enriched in one subgenome (*K*-mer frequency is >10 times that of those in other two subgenomes) were regarded as informative *K*-mers to distinguish subgenomes. The chromosomes and informative *K*-mers were grouped using hierarchical clustering. The assemblies of centromeric and telomeric regions indicate the reliability of the assembly. Tandem repeat finder (v4.09)[74] was used to identify tandem repeats along the chromosomes. To identify the centromeric regions, the largest repeat arrays composed of 155-bp units were identified and clustered. The telomeric repeats were identified by searching chromosome ends for short high-copy number repeats with the monomer 'AAACCCT'. The accuracy of subgenome distinguishing in *E. oryzicola* and *E. crus-galli* was validated by subgenome-specific *K*-mers and transposons. The centromeric regions were also identified on each chromosome of the *E. oryzicola* and *E. crus-galli* assemblies.

**Genome annotation.** The repeat elements were annotated using approaches described previously[9]. A hybrid strategy integrating ab initio predictions by Fgenesh[75] and AUGUSTUS (v3.2.2)[76], homoeologous gene evidence, and transcriptomic supports (see RNA-Seq data details in Supplementary Table 3) was applied for gene prediction. EVidenceModeler (v1.1.1) was used to integrate the gene models predicted by the above approaches to obtain a non-redundant consensus gene set[77]. Gene models were identified as those supported by homoeologous genes or transcript evidence or by at least two ab initio methods. High-confidence gene models were further filtered to remove short gene models (less than 50 amino acids) and gene models with homology to sequences in the Repbase (E value ≤ $1 \times 10^{-5}$,

identity ≥30%, coverage ≥25%). Functional annotations of protein-coding genes were conducted based on Pfam protein domains using InterProScan (v5.24–63.0)[78].

**Divergence time estimation.** Synteny among *Echinochloa* subgenomes and other species (*O. sativa*, *S. bicolor*, *D. exilis*, *S. italica*, and *P. hallii*) was first identified by pairwise BLASTP (e-value less than 1e-10) and DAGchainer (neighboring genes in a single chain to be no more than 10 genes apart, and the minimum chain length equal to five colinear genes)[79]. The non-synonymous substitution ($K_s$) values for syntenic homoeologous gene pairs were calculated by KaKs_calculator in the NG model[80]. The of $K_s$ distribution peaks were used to calculate the divergence time using the formula $T = K_s/2r$, where $r$ is the nucleotide substitution rate (here $r = 6.5 \times 10^{-9}$ mutations × $bp^{-1}$ × $generation^{-1}$)[81]. We employed the divergence of transposon sequences to infer the polyploidization time in *Echinochloa*[10].

**Phylogenetic analysis.** The occurrence of homeologous exchanges (HEs) in polyploids will affect phylogeny inferences. We identified the candidate HE regions in *E. crus-galli* genome and genes within HE regions were excluded in phylogeny analysis. About 50× short reads generated from *E. haploclada* and *E. oryzicola* assemblies were mapped against reference genome of *E. crus-galli* using Bowtie2 with default parameters[73]. Mapping depths were calculated for 100-kb sliding windows across *E. crus-galli* genome. For subgenomes A and B, the windows with mapping depth more than 30× from *E. haploclada* reads and less than 20× from *E. oryzicola* reads were regarded as candidate HE windows. The same criteria were applied for subgenome C. Neighboring candidate HE windows were merged and finally genes within 24 candidate HE regions in *E. crus-galli* were removed in following phylogeny analysis. We built the phylogeny among *Echinochloa* and other species via three methods. First, considering that $K_s$ indicates genetic divergence, we constructed the phylogeny among *Echinochloa* and neighboring species according to the pairwise $K_s$ peaks. Second, we used a concatenated matrix of 3557 single-copy orthologs identified by OrthoFinder[82] to build an ML tree. None of these single-copy genes were within candidate HE regions, which eliminated the noise by HE in the phylogeny inference within *Echinochloa*. The amino acid sequences were aligned using MAFFT (v7.310)[83]. GBlocks (v0.91b)[84] was used to trim alignment gaps with the parameter of '-b4 = 5 -b5=h'. The tree was built by using RaxML[85] with 1000 bootstrap replications. The substitution model was set as PROTGAMMA + JTT + F, as recommended by ProtTest[86] and ModelFinder implemented in IQ-TREE (v1.6.12)[87]. Third, we built a total of 3557 single gene trees individually using IQ-TREE with the best amino acid substitution model and obtained an integrated coalescent-based phylogeny by using ASTRAL (v5.7.4)[88]. To analyze the potential ILS in *Echinochloa* and introgression among genera, we first counted the numbers of three topologies in each triplet with an outgroup. Subsequently, we employed QuIBL[32], a method based on internal branch length, to quantify the proportions of ILS and introgression contributions to topology incongruence.

**Global collection and phenotyping of *Echinochloa* plants.** A total of 737 *Echinochloa* samples were collected from 16 rice-producing countries (China, Japan, Korea, Malaysia, India, and Pakistan in Asia; Brazil, US, Costa Rica, Uruguay, and Venezuela in America; Italy, Germany, Portugal, and Russia in Europe; Australia in Oceania). These samples included 328 accessions from Yangtze River basin, China, collected in the previous study[6] (Supplementary Data 3). These materials (except Brazil samples) were planted and phenotyped in the paddy fields at the China National Rice Research Institute, China, with eight individuals per accession in 2017, 2018, and 2020. The herbicide resistance phenotypes of barnyard grasses were categorized as Brazilian and Italian populations (Supplementary Note 1). The distribution of samples was illustrated using R (v4.0.3, https://www.r-project.org/).

**Resequencing and variant calling.** DNA was extracted from fresh leaves using a routine protocol. Resequencing paired-end reads were generated using Illumina HiSeq 4000. Raw data were first filtered by NGSQC Toolkit (v2.3.348)[89]. Clean paired-end reads of each accession were then aligned to the updated reference genome of *E. crus-galli* (STB08) using Bowtie2 with default parameters[72]. An integrated pipeline[6] was used in calling and filtering whole-genome variants (SNPs and InDels). To eliminate the mapping errors caused by high sequence similarities between subgenomes, the resequencing reads of STB08 were remapped to the reference, and variants were called in the same pipeline. These variants called in STB08 were removed from the final variant dataset. These variants were further filtered with the minor allele frequency (MAF) greater than 0.01 and missing rate less than 30%. For *E. colona* accessions, the clean reads were remapped to the reference genome of *E. colona* (PI473783) and variants were called using the same pipeline as mapping to *E. crus-galli* reference genome. The variants with MAF greater than 0.05 and missing rate less than 20% were further filtered. All variants were annotated by SnpEff (v3.652)[90].

**Population structure and genetic diversity.** To determine the species of each accession, read mapping rates to STB08 and genome coverage were calculated using Samtools (v1.5)[91]. FastStructure[92] was applied to infer the ancestry of all accessions with $K$-values ranging from 2 to 7, using whole-subgenome SNPs and synonymous SNPs in subgenomes A, B, and C. An approximately ML tree was

built using SNPs from both subgenomes A and B for 705 accessions (*E. colona* was excluded due to low genome coverage) with *E. walteri* as an outgroup by FastTreeMP[93], with 1000 bootstrap replications. For *E. colona* phylogeny, we constructed an approximately ML tree using whole-genome SNPs across 32 accessions of *E. colona* with one accession from *E. crus-galli* as an outgroup with 1000 bootstrap replications. These trees were illustrated in iTOL (http://itol.embl.de). A principal component analysis was performed using the PLINK software (v1.90b6.20)[94] with a pruned subset of whole-genome SNPs based on LD (10 SNPs in each 50-kb sliding window with pairwise Pearson's correlation efficient $r^2$ less than 0.5). The nucleotide diversity ($\pi$) of each population was calculated using VCFtools (v0.1.17)[95] in non-overlapping 20-kb windows. The average $\pi$ of all sliding windows was regarded as the value at the whole (sub)genome level across different groups or subgenomes.

**Demographic history.** To recover the divergence history and explore the role of gene flow in the formation of *E. crus-galli* complex, composite ML inference, based on site frequency spectrum (SFS), was conducted. To calculate the unfolded joint SFS[96,97], we used the est-sfs program (v2.03)[98] to infer the ancestral state for each allele with one individual (PI477956) from *E. walteri* as an outgroup. The joint SFS was built using only unlinked synonymous SNPs with known ancestral alleles, using the software easySFS.py (https://github.com/isaacovercast/easySFS). The SNP data of each group were down-projected to an SFS with equivalent sampling sizes across groups to decrease the effect of different levels of missing data between groups.

The divergence scenarios for the five *E. crus-galli* varieties with *E. crus-galli* var. *oryzicola* as an outgroup were set with a focus on the ancestry of *E. crus-galli* var. *crus-pavonis* (whether hybridization or continuous gene flow) (Supplementary Fig. 24). We calculated the likelihood function for different demographic scenarios using fastsimcoal2 software[34]. For each scenario, 100 000 coalescent simulations per likelihood estimation (i.e. -n100000) and 40 expectation-conditional maximization (ECM) cycles (-L40) were used as the command line parameters for each run. The Akaike information criterion (AIC) was used to compare different models. In this case, $AIC = 2k–2\ln(MaxEstLhood)$, where $k$ is the number of parameters estimated by each model, and $MaxEstLhood$ is the ML function value for each model. Moreover, when searching for an ML value, fastsimcoal2 may reach a local optimum instead of a global optimum. Thus, we repeated each step at least twice to ensure the searching did not stop in a local optimum, thereby getting better estimates of the global optimum.

The existence and direction of the gene flow between groups were tested using 14 independent scenarios (Supplementary Fig. 25). The best scenario of gene flow for 15 group pairs were then used to set migration matrix for the best scenario of group divergence. The scenario combining divergence history and gene flow were run 10 times to obtain the best parameter estimation. Thereafter, 100 independent DNA polymorphism datasets were simulated as joint SFSs conditional on estimated demographic parameters. An ML analysis was then applied to each joint SFS over 40 ECM cycles to obtain confidence intervals (CIs) for final estimates.

In addition, Stairway Plot (v2.0)[36] was used to infer effective population size change through time in recent thousand years based on both folded and unfolded SFSs. The SFS for each group was generated and folded from the same SNP dataset used for the fastsimcoal2 analysis.

**Phylogeny of *Echinochloa* chloroplast genomes.** Taken the chloroplast genome of *Panicum virgatum* (NC_015990.1) as a seed, the reads which could be used for genome de novo assembly were picked out from clean resequencing paired-end reads using NOVOplasty (v4.2)[99] and then were aligned to the reference chloroplast genome of *E. crus-galli* (BTS02; KJ000047). Using the GATK pipeline, a final set of 729 chloroplast SNPs were called across a total of 607 *Echinochloa* individuals with the filtering parameters of MAF greater than 0.05 and missing rate less than 0.2. An ML tree was built using MEGACC (v10.1.8)[100] with 1000 bootstrap replications and, the tree was rooted using the chloroplast genome of *Setaria italica* as an outgroup.

**ABBA-BABA statistics.** Incongruence between nuclear and chloroplast genome phylogeny revealed chloroplast capture in *Echinochloa*. We performed ABBA-BABA test to detect the signals of introgression from *E. oryzicola* var. *oryzicola* to *E. crus-galli* var. *crus-galli*. The test ABBA-BABA or D statistics tested for gene introgression between a triplet of three close populations 'P1', 'P2', and 'P3', using an outgroup 'O' following the topology (((P1, P2), P3), O) (Supplementary Fig. 28a). In our analysis, four topologies were tested, with *E. walteri* or *E. oryzicola* var. *hainanensis* as an outgroup (Supplementary Fig. 28b). The $f_d$ statistics was calculated in sliding windows with a window size of 100 kb and a step size of 20 kb, using the python scripts available at https://github.com/simonhmartin/genomics_general. According to the filtering criteria used by Zhou et al.[37], we converted the $f_d$ values to zero, when they were less than 0 or greater than 1. In summary, PGI (Proportion of Genomic Introgression) was used to quantify the overall introgression extent across the whole genome[37].

**Local adaptation analysis.** Population fixation statistics ($F_{st}$) were estimated using VCFtools in 20-kb sliding windows between *E. crus-galli* var. *crus-galli* and *E. crus-galli* var. *praticola* and between high-latitude and low-latitude groups in

*E. oryzicola* var. *oryzicola*. Genes in highly differentiated blocks (top 5% windows in $F_{st}$) were annotated by rice function-validated orthologs (Supplementary Data 5 and 6). Neutral test (Tajima's *D*) was calculated by VCFtools in 20-kb windows in each population. The GWAS for the FT of 461 individuals of *E. crus-galli* in 2017 was conducted using mixed linear modeling implemented in EMMAX[101]. The genome-wide significance thresholds were set as a Bonferroni correction of α = 0.05. The haplotype analyses of *Hd1* and *NRT1.1B* were carried using custom scripts. The Fisher's exact test was used to calculate significance in the GO enrichment analysis.

**Herbicide resistance analyses**. The genes encoding herbicide-target enzymes ALS, EPSPS, ACCase, and psbA in *Echinochloa* genomes were identified by BLASTP. The *Echinochloa* protein sequences of these enzymes were aligned with the corresponding sequences from the other species (*O. sativa* and *S. italica*) using MAFFT. We determined the atlas of causative mutations in global barnyard grass. The herbicide resistance of accessions sampled from Brazil and Italy were phenotyped (Supplementary Data 9 and Supplementary Note 5). For accessions with no causative mutations but herbicide resistance phenotype in a Brazilian population, we explored the possibility of gene copy number alterations underlying resistance by measuring the gene family sizes inducing non-target-sites resistance (cytochrome P450 monooxygenase, GST, and ABC transporter).

**Genomic features of barnyard millet domestication**. The genetic diversity ($\pi$) and Tajima's *D* of barnyard millet and grass and genomic differentiation ($F_{st}$) between them were estimated by VCFtools in 20-kb windows. PopLDdecay (v3.31; https://github.com/BGI-shenzhen/PopLDdecay) was used to calculate the linkage disequilibrium (LD) on variant pairs within a 300-kb window. LD decay was estimated using the distance where the Pearson's correlation efficient ($r^2$) decreased to half of its maximum value.

**Detection of CNV of the NB-ARC gene family**. Pipeline CNVings (Copy Number Variation inferred from next-generation sequencing) was used to compare gene family copy number differences at the population level using the NGS resequencing data (See details in Supplementary Note 6). CNVings estimated the copy number of NB-ARC gene family by measuring the relative abundance compared to the reference genome. The physical positions of all NB-ARC genes in the reference genome were annotated by InterProScan (*e*-value < 1e−10). The abundance of NB-ARC gene family across the whole genome was measured by the sum of the average depths of all NB-ARC loci using Mosdepth (v0.2.9)[102]. The copy number was then standardized by the average mapping depth across the whole genome. Five platinum-standard rice genomes were used to test the reliability of this pipeline (Supplementary Note 6). Whole-genome resequencing Illumina paired-end reads of wild and cultivated accessions in rice, sorghum, coix, and barnyard grass were used to quantify the size change of the NB-ARC gene family caused by domestication (Supplementary Note 6). Accessions with sequencing depth less than 10× were removed.

**Detection of genes showing high differentiation between weeds and millets**. Key genes underlying crucial domestication syndromes (e.g., shattering, plant architecture, and grain size) were identified in *E. crus-galli* and *E. colona* genomes by BLASTP to rice genes with an *e*-value less than 1e−30 and identity greater than 50%. The domains were further checked using InterProScan. In rice, shattering-related genes, including *SH1*, *sh-h*, *sh4*, *qSH1*, *NPC1*, *GRF4*, *SSH1*, *SH5*, and *SHAT1*, function in the development of the abscission zone of spikelets, whereas genes regulating the alteration of plant architecture, including *LA1*, *LPA1*, *PROG1*, *WOX6*, *WOX11*, *Hsfa2d*, *PID*, and *BRXL4*, mainly function in the gravity response pathway mediated by auxin asymmetric distribution. Grain size and weight are regulated by multiple genes, including *GS3*, *GS5*, *GS6*, *GW5*, *GW6*, *GW7*, *TGW6*, *qGW8*, *GSK2*, and other important genes (www.ricedata.cn). Non-synonymous variations with allele frequency difference greater than 0.6 between cultivated and weedy populations were regarded as candidate variations selected in domestication (Supplementary Data 11). Genome-wide differentiation between *E. crus-galli* var. *praticola* and var. *esculenta*, var. *oryzoides*, between wild and cultivated *E. colona*, was scanned by $F_{st}$. The genes within top 5% differentiated genomic regions were listed in Supplementary Data 12–14.

**Reporting summary**. Further information on research design is available in the Nature Research Reporting Summary linked to this article.

## Data availability

The *Echinochloa* genome assemblies and annotations generated in this study are available at National Genomics Data Center (NGDC) database (https://bigd.big.ac.cn) under the accession number PRJCA003883. The HiFi sequencing, Hi-C, and RNA-Seq data for genome assembly and annotation in this study have been deposited in NGDC (accession number PRJCA003883). The raw data of 409 newly re-sequenced individuals are available at NGDC (project accession PRJCA003883). The data of 328 previously re-sequenced barnyard grass accessions were retrieved from NGDC (accession number PRJCA001519). Besides *Echinochloa* genomes, other 36 monocot genomes used in this study were downloaded from EnsemblPlants (https://plants.ensembl.org) or National Center for Biotechnology Information (NCBI) GenBank (https://www.ncbi.nlm.nih.gov/) (Supplementary Data 1). The resequencing data of rice, sorghum, and Job's tears were downloaded from the European Nucleotide Archive (ENA) (https://www.ebi.ac.uk/ena/browser/home) (Supplementary Data 10). Source data are provided with this paper.

## Code availability

The custom scripts used in this study have been deposited in the GitHub repository [https://github.com/bioinplant/Echinochloa_genome].

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

## Acknowledgements

This project was supported by the Department of Science and Technology of Zhejiang Province (2022C02032 and 2020C02002), the National Natural Science Foundation (32170621 and 31901899), Zhejiang Natural Science Foundation (LZ17C130001), Jiangsu Collaborative Innovation Center for Modern Crop Production and 111 Project (B17039) to L.F. We thank Yongfeng Li (Institute of Plant Protection, Jiangsu Academy of Agricultural Sciences, China) for help with material collection and Prof. Ya-Long Guo (Institute of Botany, Chinese Academy of Sciences, China) for his insightful suggestions.

## Author contributions

L.F. and C.-Y.Y. designed and supervised the study. W.T., A.M.J., L.S., K.M.O., B.-K.S., T.-S.C, D.W., H.Y., and C.-Y.Y. collected samples. W.T., C.-Y.Y., D.W., L.J., B.J., X.W., C.D., and L.G. phenotyped samples and prepared materials for sequencing. L.C., A.M.J., and L.S. conducted the herbicide resistance experiments. D.W., L.J., B.J., and C.D. performed genome assembly and annotation. D.W., E.S., C.-Y.Y., Y.F., S.L., H.-Y.L., F.L., B.J., L.X., H.X., and Q.Q. performed comparative genomic and population genetics analyses. D.W., L.F., C.-Y.Y., Q.-H.Z., and Q.Y. wrote the manuscript. K.M.O., Y.Q., H.Y., M.P.T., H.C., S.D. and J.Q. discussed the results and edited the manuscript. All authors read and approved the manuscript.

## Competing interests

The authors declare no competing interests.
