## [Peer Review File · Nature Communications]

Genomic Insights into the Evolution of Echinochloa Species as Weed and Orphan CropReviewers' Comments:

Reviewer #1:

Remarks to the Author:

In this manuscript, Wu et al. created, improved and analyzed extensive genomic resources for barnyard grasses (*Echinochloa* spp.). Some *Echinochloa* species are devastating weeds in agriculture, while others are cultivated and serve as food sources. Analyzing the population structure of *Echinochloa* is thus important to understand the evolution of weediness and domestication in this genus. The authors created chromosome-scale genome assemblies of three polyploidy *Echinochloa* species and re-sequenced a diversity panel of 737 accessions. While several near complete *Echinochloa* assemblies have been generated previously, this manuscript presents the first chromosome-level assemblies of polyploidy species of this genus. While most of the findings presented here are not entirely novel (e.g., a loss of NLR diversity in *Echinochloa* has been reported previously), the extent of the genomic resources generated here and the scope of the analyses are impressive. In general, the manuscript is very clear and very well written. I do have a few comments/questions that the authors should address:

- The authors developed a neat approach named 'diploid-assisted scaffolding (DipHiC)' to assign contigs to the various sub-genomes of polyploidy *Echinochloa* species. The authors use a sequence-based approach (k-mers) for validation. However, I miss a truly independent validation of this scaffolding method, for example by using a genetic map or chromosome painting. I assume that large non-homoeologous translocations (such as the translocation between chromosome 4A and 7B in hexaploid wheat) would be missed with this approach. I am aware that the construction of a genetic map or the establishment of chromosome painting is tedious and might not be within the scope of this manuscript, but the authors should clearly point out this limitation (if it is one) in the main text. I am also not sure if it is correct to refer to 'chromosome-level' assemblies without an independent validation.
- The gene annotation is only poorly described. Line 856 states that RNA-Seq data was used for gene annotation. Were RNA-Seq data produced for all three species and were the tissues and sequencing depth comparable? How was the filtering for high-confidence and low-confidence genes done? How many low-confidence genes are there?

Minor comments:

- Line 105: 'have the most compact hexaploid genomes currently known in plants'?
- Line 152: What are "clean" PacBio HiFi reads?
- Line 176: How many of the core conserved Poales genes were present in single copy and multiple copies in the *E. colona* genome?
- Line 448: The authors should also cite this paper when mentioning interspecific introgressions from wheat (<https://www.nature.com/articles/s41586-020-2961-x>).

Reviewer #2:

Remarks to the Author:

The authors provides two high quality barnyard grasses reference genomes and sequences of 737 individuals. Overall, this work is excellent.

One of the novelties in this manuscript is that the authors proposed a new pipeline (DipHiC) to phasing and scaffolding of subgenomes in an allo-tetraploid species *E. oryzicola* and two allo-hexaploid species *E. crus-galli* and *E. colona*. DipHiC is able to separate subgenomes by utilizing the mapping depth of a diploid close relatives and mutually exclusive relationship between contigs from the target polyploid genome. Assessment using Hi-C contact heatmap (e.g. SFigure 4) and distribution patterns transposon elements (SFigure 6) showed that the genome assemblies are of high-quality. However, requiring additional information (for instance, a diploid assembly and 20-kb mate pair library) and a

less robust pipeline may limit the widespread use of the strategy. In fact, separating the subgenomes of allo-polyploid species is less complicated than phasing auto-polyploid genomes especially for those with high divergence between subgenomes. Evidences show that the repetitive DNA sequences are rapidly evolving in subgenomes, which have been used to distinguish subgenomes in a couple of allopolyploid species, e.g. *Miscanthus*. In addition, an unsupervised partitioning method (polycracker) was proposed to achieve this goal without any prior knowledge, and had been successfully applied on two allopolyploid species, tobacco and wheat. Given the previous knowledge, I question the intention and necessity to develop such a complex pipeline to assemble these genomes.

Another major concern is that the authors frequently mix results and discussion, making their conclusion unreliable sometimes. For instance, Lines 462-463, "Either sexual hybridization or asexual grafting can lead to chloroplast capture or horizontal transfer". Are they making conclusion or just citing a previous publication to support their conclusion? At least a reference needed here. In addition, Lines 409-428, the whole paragraph should be moved to discussion part rather than presented in Results.

Lines 213-214, "Compared with the diploid *E. haploclada* (37510 genes), the polyploid subgenomes showed significant gene losses". This conclusion is not obvious.

In Line 230, the authors claimed that "NB-ARC genes were significantly enriched on chromosome 4", but did not provide any statistics test. The same issue should be applied whenever "significant/significantly" are mentioned, e.g. Lines 447, 525, 544 etc. In addition, SFigure 10 should be normalized using z-score.

Lines 287-294, the authors concluded that the discordance between gene and species trees in *Echinochloa* was more likely caused by ILS rather than introgression by presenting two evidences: 1) 1:1:1 ratio in the gene counts of the three trees and 2) QuIBL results. But they did not provide details for explanation. Why the 1:1:1 ratio could indicate ILS and what QuIBL results lead authors to make such conclusion?

Lines 840-850, how the accuracy was validated by specific-kmer?

There are many PERL scripts on github, but without enough description, please add a rich README.md.

Reviewer #3:

Remarks to the Author:

Synopsis: Researchers sequenced the genomes of three *Echinochloa* species: hexaploids *E. crus-galli* and *E. colona* and tetraploid *E. oryzicola*. Only one *Echinochloa* species has been previously sequenced, the diploid *E. haploclada*. The authors then sequenced hundreds of new *Echinochloa* plants from around the world to add to their panel of 737 *Echinochloa* collections. Using the new genomes and diversity panel the authors set out to investigate questions around *Echinochloa* genus' genome biology, evolution, phylogeny, domestication, gene transfer, and abiotic stress resistance especially herbicides.

General Comments: This manuscript represents a staggering amount of work and explores the *Echinochloa* genomes in several ways. It is so much and so dense that this might be to its detriment. The sheer number of different topics and questions being addressed means the central through line is easily lost. Maybe it would be better for the manuscript to be broken into a number of smaller sequential manuscripts that address each topic in turn and fully explores their individual hypotheses and nuances. I really enjoyed that the manuscript addresses several outstanding questions and hypotheses previously posited, while opening the door for new avenues of research and discovery. The writing is excellent and little needs to be edited.

I have two large concerns. The first is the rigor that was applied when making when making the phylogenetic inferences and the conclusions about introgression versus ILS. I bring up several concerns and questions for that portion of the manuscript that should be considered. The second major concern is the selection of figures. They seem to serve to impress and overwhelm the reader rather than tell a consistent and convincing narrative. Careful production and selection of figures is recommend to streamline the narrative and tell a more compelling overarching story.

Line Edits

Line 82: The use of “inevitably” implies a strong causal link, while the sentiment is appreciated perhaps herbicide tolerance in all use cases does not inevitably lead to herbicide resistance.

Line 92: “Evidently, barnyard grasses are excellent resources or systems for understanding plant stress”. This abrupt transition could be made clearer that this. Is it-because of their success in the face various abiotic stresses?

Line 214: “significant gene losses.” The word significant is being used without obvious statistical method, were losses determined using CAFÉ?

Line 220: Barabaschi et al. 2020 posits a fitness cost but makes no such calculation. Rather these authors infer it based on genomic comparisons. Similarly, Ye et al. 2020 do not seem to measure fitness cost but rather gene expansion. Recommend that authors cite Review of fitness cost by Brown and Rant 2013, or other similar sources used in Ye et al. 2020 to first establish the strong link between R genes and fitness cost.

Line 228 and 230: “significantly” is used again without obvious statistical method

Lines 303-323: The authors set out to establish if discordance between predicted discordance times and k_s estimates are driven by incomplete lineage sorting and/or introgression. The authors calculated k_s divergence using a COGE like approach, if not COGE exactly. Phylogenetic methods, were based off of Orthofinder and RAxML. Coalescent approach done with ASTRAL.

While the authors work using QuIBL is respectable, their handling of the concatenated RAxML tree is unsatisfying. They report no alignment QC, partitioning or model selection. While Orthofinder is a nice general purpose program for generating Orthogroups, it is no substitute for handling the data rigorously. Questions abound.

1. Why not use nucleotide sequences with a codon aware aligner instead of the proteins. If the species are close enough to calculate reliable k_s peaks, aligning at the nucleotide level is a tractable task.
2. Since whole genome sequences are being used, why not employ the same set of homologous genes identified via synteny in genome guided orthology approach? This would be even more reliable than orthofinder since it incorporates genomic position, and basic data handling would allow for the use of amino acid or nucleotide sequences as authors saw fit.

In the case where Orthofinder is employed. I caution the authors that past experience with Orthofinder strongly suggests that the data will only benefit from a more rigorous handling of the sequences. This is because the alignments are frequently gappy and poorly aligned under alignment settings provided by most automated aligners and thus a QC step is required. The QC step and post processing should address the following.

1. How were gaps dealt with, the introduction of gaps into multiple sequence alignments can have a severe consequences.
2. Was GBlocks or a similar program employed to limit the amount of noise in the alignment? If so why was it not reported and what were the flags used to ensure reliable alignments were used in the concatenated sequence

3. Was partitionfinder2 or some suitable substitute used to ensure that the correct models were applied throughout the concatenated matrix?
4. What model was used for RAXML?

The same point about alignment QC applies to the Multiple Sequence Alignments (MSAs) used for IQTree. IQtree does an excellent job testing for best substitution models, however it does not manage gapped or poorly aligned sequenced output regularly produced by MAFFT with default settings. This is particularly problematic for sequences with large disordered domains and for orthogroups in which one or more sequences contains novel amino acid sequence insertions. These must be dealt with before MSAs are given to IQTree. Again, there is no obvious evidence that this was done. Poor alignments produce unreliable inferences.

The authors have not convinced me that they have handled the analysis rigorously enough to support the claims based on this analysis. In particular, it seems just as likely that they are measuring artifact introduced by the handling of data as they are measuring ILS or introgression. I admit that measuring ILS and introgression is a worthy undertaking, but has not been convinced by the current handling of the data.

Ln574: Please cite "Tranel, Patrick J., and Terry R. Wright. "Resistance of weeds to ALS-inhibiting herbicides: what have we learned?." *Weed Science* 50, no. 6 (2002): 700-712." or "Murphy, Brent P., and Patrick J. Tranel. "Target-site mutations conferring herbicide resistance." *Plants* 8, no. 10 (2019): 382." It is an essential piece of literature addressing TSR for ALS inhibitors

Line 613: Please cite "Patterson, Eric L., Dean J. Pettinga, Karl Ravet, Paul Neve, and Todd A. Gaines. "Glyphosate resistance and EPSPS gene duplication: convergent evolution in multiple plant species." *Journal of Heredity* 109, no. 2 (2018): 117-125." It is a more focused review of the effect that CNVs have on glyphosate resistance.

Lines: 668-675 "Significantly" is used twice without an obvious statistical method. The pipeline CNVings is briefly described, but no mention is made of statistical test is applied to determine differences, perhaps it was a T-test.

Figure 1:

Line 1106-1112: This circus plot can be deconvoluted. Many of these tracks are unnecessary (e.g. GC content) as they don't increase our understanding of the narrative and are for the most part uninteresting (especially crammed in a tiny figure). Only include the data relevant to your point(s). This will highlight the more relevant tracks (like Copia, Gypsy, etc).

Line 1113: Not sure this is interesting in a graphical form. Just extra complexity and we understand well from the text how complete the genome(s) are. Also, it may allow more room for panel D and/or E to be larger and more interpretable.

Figure 2:

Line 1139-1141: The size of this figure versus the complexity of the figure are preposterous. A simple cartoon, collapsing redundant branches, would suffice to tell the admixture story you are trying to tell.

Figure 3:

Line 1161-1164: Same comment as before, what benefit do we get as a reader to seeing every branch and branch length? Simplify your figures where you can to tell a more concise and interpretable story.

Line?: The figure panel of the two clades (Below the ML phylogeny) is not described in the figure head.

Figure 5: Why are panel 1D separated so far from Figure 5 when both are discussing NB-ARC?

Shouldn't panel 1D be part of this figure instead?

Reviewer #4:

Remarks to the Author:

The article brings novel insights into an evolution and agronomic significance of high polyploid grasses. Although there are several genome assemblies of allopolyploid species already available, the comprehensive analysis presented here brings valuable general insights into allopolyploid evolution in general. The model system is fascinating and in depth investigated by an impressive collection of global diversity of the three focal species. The analyses are diverse and at many aspects very comprehensive (esp. genome assembly, investigation of range-wide diversity) while the analysis of selective footprints of local adaptation, herbicide resistance and domestication is rather limited to testing particular hypotheses, which are addressed by a diverse and somehow inconsistent approaches, that seem not to be always clearly justified. Although I do not see any major significant flaw, I have the following comments aiming to clarify and potentially improve some aspects of the study.

1) The phylogeny of *Echinochloa* species and subgenomes is crucial for further interpretations. I understand the reasons why there are no more genomes assembled and sequences but I wonder whether there are at least some sequence data for additional species available. Addition of such data (e.g. in a supplementary tree involving a subset of loci gathered also from current genomes) might better clarify the origin of the other subgenomes, esp. FH and EH that are currently isolated in the tree. Putting these newly assembled (sub)genomes into a broader context of *Echinochloa* evolution would help understanding their evolution.

2) Many recently assembled allopolyploid genomes demonstrate surprising stasis and lack of large genomic exchanges among the sub-genomes (e.g. Burns et al. 2021 Nat Ecol Evol and refs therein), however, it would be still worth checking and discussing if that is also the case for *Echinochloa* or whether they are in fact such homeologous recombinations present. Lack of homeologous exchange is an assumption of many downstream analyses but has not been discussed.

3) What is the breeding system of the species? In case of strong selfing, this shall be taken into account in the interpretations of diversity and differentiation from population resequenced data. E.g. could the decrease in nucleotide diversity among *E.c.* varieties (l. 650-653) be attributed to breeding system shift, possibly associated with domestication?

4) l 217-240 The finding of contraction of NB-ARC gene family is interesting, it is however unclear how much this case represents an outlier in gene contraction among different gene families in *Echinochloa*. Hence, it is hard to speculate on adaptive significance of such a reduction (see l. 219) and it is a bit unclear the motivation why an entire paragraph is devoted to discussion of this particular family. Is the NRC-ARC family contraction significantly higher than is a genome-wide average in *Echinochloa*?

5) Fig. 1E: It is totally unclear from where the time estimates come from. Are they based on some uniform substitution rate estimate in grasses? If so, such an assumption seem to be tricky given known variation in substitution rates, and surely presenting a single point estimate of divergence time, based on such an approach, may be very misleading. It will be beneficial either to date the splits based on calibrated phylogeny (including confidence intervals for each node), or such numbers shall be excluded from the main figure to avoid potential misleading divergence times (these values seem not to be very important for further interpretation anyway). Related to that, a phylogeny based on multispecies coalescent reconstruction across multiple loci (i.e. Astral, Fig S13), i.e. an approach directly developed for phylogeny reconstructions, may provide more robust than the relatedness based on the distribution of nonsynonymous substitutions.

6) The motivation for using particular methods for detection selection is not always clear and raises concerns about the homogeneity of the approaches throughout the ms. Specifically, why results of flowering-time variation are based on a genome-wide divergence (F_{st}) outlier scan while e.g. the domestication process is investigated by allele frequency spectra-oriented approach targeting only a subset of (pre-selected?) loci?

7) In some candidate loci it is unclear how the variation with these loci looks in the other two homeologs within a hexaploid (e.g. Hd1 is reported only from subgenome AH but nothing is mentioned about subgenome BH and CH). Are they pseudogenes? Invariable but likely functional loci? Variable but not enough to be considered outliers? Undetected? Such a context would provide important hints about the role of allopolyploidisation and potential sub/neofunctionalisation in adaptation.

8) The entire section on herbicide resistance (l. 569-627) lacks a primary exploratory genome-wide approach and/or justification why the entire part focuses only on a small subset of particular loci. As a result, it appears as somehow selective "cherry-picking" of certain loci without a justification how much variation in such loci is exceptional in the context of the entire genome. Thus, although this section provides some hints of potentially valuable loci for a follow-up it does not adhere to the major focus of the paper on genome-wide variation and evolutionary significance of genomic variation related to allopolyploidy. At current state this section provides rather some particular supplementary information targeted to a very specialised audience.

9) The Discussion is very brief and descriptive. Although not being an expert in the field of agricultural genomics, I doubt there is so scarce literature on the topics as it is presented here (in total four citations in the entire Discussion section). E.g. considering additional well-researched cases of the evolution of agricultural weeds such as *Amaranthus tuberculatus* (Kreiner et al. 2019 PNAS, Kreiner et al. 2018 ARPB and refs. therein) would be helpful. Also putting the result into a context of other emerging studies on genomic evolution in allopolyploids (e.g. *Arabidopsis suecica*, *Brassica napus*, *Gossypium*, ...) might increase the impact of the study.

10) Although the authors at multiple places highlight the significance of their study for follow-up genetic studies, list of candidate loci for adaptation towards the discussed factors (resistance, latitude, domestication...) are missing. It would be very helpful providing complete lists of candidate loci found genome wide by F_{st} scans, GWAS approaches and SFS-based scans as such primary data would provide molecular biologist a key resource for addressing particular genes in this emerging model.

Minors

- I consider crucial the fact the studied group is ALLOpolyploid. I suggest adding this information already to the abstract.
- l. 105-106 maybe "most" missing from "the MOST compact genome"
- l. 253 "calculated the pairwise K_s " is a jargon. Please explain the principle/motivation behind this analysis.
- l. 321 based on what the authors conclude that "introgression may have facilitated an increase in environmental adaptation"? This seems rather speculative given the presented K_s analysis.
- l. 338 what is the ploidy of *E. walteri* that is used as an outgroup in the resequencing data?
- l. 397 unclear, which measure the % of "admixture" are based on
- l. 506 unclear what are the "outliers", this shall be also explained in the text not only in the Fig caption
- l. 536 how many individuals were used for the GWAS? How many loci others than Hd1 were found to be significantly associated with FT?

REVIEWER COMMENTS

Reviewer #1 (Remarks to the Author):

In this manuscript, Wu et al. created, improved and analyzed extensive genomic resources for barnyard grasses (*Echinochloa* spp.). Some *Echinochloa* species are devastating weeds in agriculture, while others are cultivated and serve as food sources. Analyzing the population structure of *Echinochloa* is thus important to understand the evolution of weediness and domestication in this genus. The authors created chromosome-scale genome assemblies of three polyploidy *Echinochloa* species and re-sequenced a diversity panel of 737 accessions. While several near complete *Echinochloa* assemblies have been generated previously, this manuscript presents the first chromosome-level assemblies of polyploidy species of this genus. While most of the findings presented here are not entirely novel (e.g., a loss of NLR diversity in *Echinochloa* has been reported previously), the extent of the genomic resources generated here and the scope of the analyses are impressive. In general, the manuscript is very clear and very well written. I do have a few comments/questions that the authors should address:

Response:

Thanks for the positive comments on our manuscript.

- The authors developed a neat approach named ‘diploid-assisted scaffolding (DipHiC)’ to assign contigs to the various sub-genomes of polyploidy *Echinochloa* species. The authors use a sequence-based approach (k-mers) for validation. However, I miss a truly independent validation of this scaffolding method, for example by using a genetic map or chromosome painting. I assume that large non-homoeologous translocations (such as the translocation between chromosome 4A and 7B in hexaploid wheat) would be missed with this approach. I am aware that the construction of a genetic map or the establishment of chromosome painting is tedious and might not be within the scope of this manuscript, but the authors should clearly point out this limitation (if it is one) in the main text. I am also not sure if it is correct to refer to ‘chromosome-level’ assemblies without an independent validation.

Response:

Thanks for your comments. *K*-mer based clustering of contigs is one of the approaches for clustering. In addition, we also considered contig allelism and inter-contig linkages, which could eliminate the faults caused by using only *K*-mer based clustering method. Nevertheless, additional independent evidence to validate the assembly accuracy is needed. The assembly of polyploid genome is still a challenge, due to high sequence similarities between subgenomes and homeologous exchanges, which requires more evidence (e.g. genetic maps, optical maps) to validate and improve

the assembly accuracy and completeness. We have added some discussion on the concern.

For the term “chromosome-level”, it is now commonly used in most genome papers, in which genomes were assembled by the next-/ third-generation sequencing and HiC data with no validation of genetic maps, such as the two Nature Communications papers just published on November 26th (Ma et al., Nat Commun 12, 6929 (2021); Guo et al., Nat Commun 12, 6930 (2021)).

Guo, X., Fang, D., Sahu, S.K. et al. *Chloranthus* genome provides insights into the early diversification of angiosperms. Nat Commun 12, 6930 (2021).

<https://doi.org/10.1038/s41467-021-26922-4>

Ma, J., Sun, P., Wang, D. et al. The *Chloranthus sessilifolius* genome provides insight into early diversification of angiosperms. Nat Commun 12, 6929 (2021).

<https://doi.org/10.1038/s41467-021-26931-3>

- The gene annotation is only poorly described. Line 856 states that RNA-Seq data was used for gene annotation. Were RNA-Seq data produced for all three species and were the tissues and sequencing depth comparable? How was the filtering for high-confidence and low-confidence genes done? How many low-confidence genes are there?

Response:

Thanks for your suggestion. We have added more detailed description about gene annotation. The RNA-seq datasets for genome annotation have been added in Supplementary Table 18 and Method part. All gene structures predicted by *ab initio* predictions, homeologous gene evidence and transcriptomic support (RNA-seq), were integrated into consensus gene models by EVIDENCEModeler. Gene models were identified as those supported by homeologous genes or transcript evidence or at least two *ab initio* methods. High-confidence gene models were further filtered to remove short gene models (less than 50 amino acids) and those with homology to sequences in the Repbase (E value $\leq 1e-5$, identity $\geq 30\%$, coverage $\geq 25\%$).

Species	Low-confidence models to be removed		High-confidence
	Invalid length	Repeat	
E.crus-galli	5135	102919	104780
E.colona	4613	78488	93697
E.oryzicola	3181	75954	68544

Minor comments:

- Line 105: ‘have the most compact hexaploid genomes currently known in plants’?

Response:

We have corrected it. The genomes of two hexaploid *Echinochloa* species (*E. crus-galli* and *E. colona*) are among the most compact hexaploid genomes currently known in plants.

- Line 152: What are “clean” PacBio HiFi reads?

Response:

We have deleted it. It should be “consensus reads generated from Pacbio CCS”.

- Line 176: How many of the core conserved Poales genes were present in single copy and multiple copies in the *E. colona* genome?

Response:

Of the all 4896 BUSCOs (poales_odb10), 4747 (97.0%) were completely covered, including 592 (12.1%) single and 4155 (84.9%) duplicate. 23 (0.5%) were fragmented and 126 (2.5%) were missing.

- Line 448: The authors should also cite this paper when mentioning interspecific introgressions from wheat (<https://www.nature.com/articles/s41586-020-2961-x>).

Response:

We have added the paper of Wheat Pangenome (Walkowiak et al., 2021, Nature).

Reviewer #2 (Remarks to the Author):

The authors provide two high quality barnyard grasses reference genomes and sequences of 737 individuals. Overall, this work is excellent.

Response:

Thanks for the very positive comments on our manuscript.

One of the novelties in this manuscript is that the authors proposed a new pipeline (DipHiC) to phasing and scaffolding of subgenomes in an allo-tetraploid species *E. oryzicola* and two allo-hexaploid species *E. crus-galli* and *E. colona*. DipHiC is able to separate subgenomes by utilizing the mapping depth of a diploid close relatives and mutually exclusive relationship between contigs from the target polyploid genome. Assessment using Hi-C contact heatmap (e.g. SFigure 4) and distribution

patterns transposon elements (SFigure 6) showed that the genome assemblies are of high-quality.

However, requiring additional information (for instance, a diploid assembly and 20-kb mate pair library) and a less robust pipeline may limit the widespread use of the strategy. In fact, separating the subgenomes of allo-polyploid species is less complicated than phasing auto-polyploid genomes especially for those with high divergence between subgenomes. Evidences show that the repetitive DNA sequences are rapidly evolving in subgenomes, which have been used to distinguish subgenomes in a couple of allopolyploid species, e.g. *Miscanthus*. In addition, an unsupervised partitioning method (polycracker) was proposed to achieve this goal without any prior knowledge, and had been successfully applied on two allopolyploid species, tobacco and wheat. Given the previous knowledge, I question the intention and necessity to develop such a complex pipeline to assemble these genomes.

Response:

Thanks for your comments and suggestions.

Firstly, we tested several regular HiC scaffolding software and they did not performed well in *Echinochloa* genomes, even for AllHiC, a method specially designed for polyploid genome scaffolding (Supplementary Fig. 1a), which resulted from high similarities among three subgenomes.

In the case of tetraploid *Miscanthus genome* (Mitros, et al., 2020, Nature Communications), the scaffold-level assembly was integrated into chromosome-level by using large-insert mate library, HiC and genetic map firstly. The chromosomes were then assigned into subgenomes using a *K*-mer based method. It was a strategy of “chromosome building firstly and subgenome phasing secondly”. In another *Miscanthus* genome project (Zhang et al., 2021, Nature Plants), the integration of 10X genomic data, HiC data, Bionano optical data and high-density genetic map scaffolded contigs into chromosomes, but only 91.03% sequences were anchored on chromosomes. We did attempt to scaffold contigs into chromosomes directly, but failed. The negative results were expected, because of the hexaploidy of *E. colona* and *E. crus-galli*. Thus we decided to carry another strategy of “subgenome phasing firstly and chromosome building secondly”.

As you said, there are some tools using subgenome characteristic sequences to phase subgenomes, including polyCRACKER using *K*-mer or repetitive sequences and there are successful cases in tobacco and wheat. But in *Echinochloa*, this kind of method did not work well. The principle of polyCRACKER is based on the richness of subgenome-specific *K*-mers or repeat elements, which means that only in polyploid genomes with larger divergence among subgenomes (longer time for specific repeat

elements expansion) and larger repeat element proportions, this method would perform well in subgenome phasing. In fact, the repeat elements accounted for ~80% (Xu et al., 2017, PNAS; Sierro et al., 2014, Nature communications) in tobacco and ~85% in wheat (IWGSC, 2018, Science), while only ~44%-51% in *Echinochloa* species. In addition, the contig continuity (N50 values) of *E. crus-galli* and *E. oryzicola* using CLR sequencing was limited. In our case, the genomes of ancestral parents (or relatives) of *Echinochloa* species are available, we thought it is a good way to use the ancestral diploid genome to phase subgenomes. Using ancestral diploid to phase subgenomes has been also used in other genome projects, such as octoploid strawberry (Edger et al., 2019, Nature genetics), tetraploid elephant grass (*Cenchrus purpureus*) (Yan et al., 2020), *Arabidopsis kamchatica* (Paape et al., 2018, Nature communications).

In general, several factors required us to use DipHiC. First, *E. colona* and *E. crus-galli* are hexaploid, which hindered the building of chromosomes using HiC directly by popular methods (Lachesis and AllHiC). Second, the contig continuity (N50 values) of *E. crus-galli* and *E. oryzicola* using CLR sequencing was limited. Short contigs could not be effectively assigned into subgenomes by characteristic sequences. Third, the repetitive elements were relatively low, which would not provide enough subgenome-specific sequences as markers to phase subgenome.

Mitros T, Session AM, James BT, et al. Genome biology of the paleotetraploid perennial biomass crop *Miscanthus*. Nat Commun. 2020 Oct 28;11(1):5442. doi: 10.1038/s41467-020-18923-6.

Zhang G, Ge C, Xu P, et al. The reference genome of *Miscanthus floridulus* illuminates the evolution of Saccharinae. Nat Plants. 2021 May;7(5):608-618. doi: 10.1038/s41477-021-00908-y.

Gordon SP, Levy JJ, Vogel JP. PolyCRACKER, a robust method for the unsupervised partitioning of polyploid subgenomes by signatures of repetitive DNA evolution. BMC Genomics. 2019 Jul 12;20(1):580. doi: 10.1186/s12864-019-5828-5.

Xu S, Brockmüller T, Navarro-Quezada A, et al. Wild tobacco genomes reveal the evolution of nicotine biosynthesis. Proc Natl Acad Sci U S A. 2017 Jun 6;114(23):6133-6138. doi: 10.1073/pnas.1700073114.

Sierro N, Battey JN, Ouadi S, et al. The tobacco genome sequence and its comparison with those of tomato and potato. Nat Commun. 2014 May 8;5:3833. doi: 10.1038/ncomms4833.

IWGSC,. Shifting the limits in wheat research and breeding using a fully annotated reference genome. Science. 2018 Aug 17;361(6403):eaar7191. doi: 10.1126/science.aar7191.

Edger PP, Poorten TJ, VanBuren R, et al. Origin and evolution of the octoploid strawberry genome. Nat Genet. 2019 Mar;51(3):541-547. doi: 10.1038/s41588-019-0356-4.

Yan Q, Wu F, Xu P, et al. The elephant grass (*Cenchrus purpureus*) genome provides insights into anthocyanidin accumulation and fast growth. Mol Ecol Resour. 2021 Feb;21(2):526-542. doi: 10.1111/1755-0998.13271.

Paape T, Briskine RV, Halstead-Nussloch G, et al. Patterns of polymorphism and selection in the

Another major concern is that the authors frequently mix results and discussion, making their conclusion unreliable sometimes. For instance, Lines 462-463, “Either sexual hybridization or asexual grafting can lead to chloroplast capture or horizontal transfer”. Are they making conclusion or just citing a previous publication to support their conclusion? At least a reference needed here.

In addition, Lines 409-428, the whole paragraph should be moved to discussion part rather than presented in Results.

Response:

Thanks for the suggestions. We have revised the main text and separated the results and discussion according to your suggestions.

Lines 213-214, “Compared with the diploid *E. haploclada* (37510 genes), the polyploid subgenomes showed significant gene losses”. This conclusion is not obvious.

Response:

We have deleted the word “significant”.

In Line 230, the authors claimed that “NB-ARC genes were significantly enriched on chromosome 4”, but did not provide any statistics test.

The same issue should be applied whenever “significant/significantly” are mentioned, e.g. Lines 447, 525, 544 etc.

Response:

Thanks for your suggestion. We have added the statistic tests when we mentioned “significant”.

In addition, SFigure 10 should be normalized using z-score.

Response:

We have revised it according to your suggestion (see the following figure).

Lines 287-294, the authors concluded that the discordance between gene and species trees in *Echinochloa* was more likely caused by ILS rather than introgression by presenting two evidences: 1) 1:1:1 ratio in the gene counts of the three trees and 2) QuIBL results. But they did not provide details for explanation. Why the 1:1:1 ratio could indicate ILS and what QuIBL results lead authors to make such conclusion?

Response:

Topology counting of a triplet with an outgroup is a commonly-used method to explore ILS and the 1:1:1 ratio of three topologies implied possible ILS around the divergence node (e.g., Ma et al., 2021, Nature Communications; Yang et al., 2020, Nature Plants; One Thousand Plant Transcriptomes Initiative, 2019, Nature). Output of QuIBL directly reported the proportion of genes whose topologies were influenced by ILS or introgression only (See details in Supplementary Table 2). Almost 100% of the genes used showed influences by ILS rather than introgression. Also topology distribution could be used to distinguish the pattern of ILS and introgression and this analysis has been added (see the following figure). Based on triplet topology counting, QuIBL and topology distribution, we could speculate that the observed radiative evolution in *Echinochloa* was mainly caused by ILS.

Distribution of three topologies in triplet ((FH, AH, E. haploclada), O. sativa)

Ma J, Sun P, Wang D, et al. The *Chloranthus sessilifolius* genome provides insight into early diversification of angiosperms. Nat Commun. 2021 Nov 26;12(1):6929. doi: 10.1038/s41467-021-26931-3.

One Thousand Plant Transcriptomes Initiative. One thousand plant transcriptomes and the phylogenomics of green plants. Nature. 2019 Oct;574(7780):679-685. doi: 10.1038/s41586-019-1693-2.

Yang Y, Sun P, Lv L, et al. Prickly waterlily and rigid hornwort genomes shed light on early angiosperm evolution. Nat Plants. 2020 Mar;6(3):215-222. doi: 10.1038/s41477-020-0594-6.

Lines 840-850, how the accuracy was validated by specific-kmer?

Response:

Each genome of *E. colona*, *E. oryzicola* and *E. crus-galli* was split into 13-mers. The frequencies of *K*-mers were counted for each chromosome. Only the *K*-mers mainly distributed in one subgenome (counts proportion in this subgenome greater than 95% of total counts) were kept as subgenome-specific *K*-mers and potentially could be used to phase subgenome. For each subgenome in any *Echinochloa* polyploids, there existed some subgenome-specific *K*-mers.

There are many PERL scripts on github, but without enough description, please add a rich README.md.

Response:

Thanks for your comments. We have updated the README files for scripts used in this study, particularly the pipelines used for genome scaffolding of three *Echinochloa* polyploids. See details at https://github.com/bioinplant/Echinochloa_genome.

Reviewer #3 (Remarks to the Author):

Synopsis: Researchers sequenced the genomes of three Echinochloa species: hexaploids *E. crus-galli* and *E. colona* and tetraploid *E. oryzicola*. Only one Echinochloa species has been previously sequenced, the diploid *E. haploclada*. The authors then sequenced hundreds of new Echinochloa plants from around the world to add to their panel of 737 Echinochloa collections. Using the new genomes and diversity panel the authors set out to investigate questions around Echinochloa genus' genome biology, evolution, phylogeny, domestication, gene transfer, and abiotic stress resistance especially herbicides.

General Comments: This manuscript represents a staggering amount of work and explores the Echinochloa genomes in several ways. It is so much and so dense that this might be to its detriment. The sheer number of different topics and questions being addressed means the central through line is easily lost. Maybe it would be better for the manuscript to be broken into a number of smaller sequential manuscripts that address each topic in turn and fully explores their individual hypotheses and nuances. I really enjoyed that the manuscript addresses several outstanding questions and hypotheses previously posited, while opening the door for new avenues of research and discovery. The writing is excellent and little needs to be edited.

I have two large concerns. The first is the rigor that was applied when making when making the phylogenetic inferences and the conclusions about introgression versus ILS. I bring up several concerns and questions for that portion of the manuscript that should be considered. The second major concern is the selection of figures. They seem to serve to impress and overwhelm the reader rather than tell a consistent and convincing narrative. Careful production and selection of figures is recommend to streamline the narrative and tell a more compelling overarching story.

Response:

Thanks for your positive comments on our manuscript. We have carefully addressed the questions you raised (see the following response for your concrete question).

Line Edits

Line 82: The use of “inevitably” implies a strong causal link, while the sentiment is appreciated perhaps herbicide tolerance in all use cases does not inevitably lead to herbicide resistance.

Response:

We have deleted this word.

Line 92: “Evidently, barnyard grasses are excellent resources or systems for understanding plant stress”. This abrupt transition could be made clearer that this. Is

it-because of their success in the face various abiotic stresses?

Response:

We have revised this sentence.

Line 214: “significant gene losses.” The word significant is being used without obvious statistical method, were losses determined using CAFÉ?

Response:

We have revised this sentence by deleting “significant”. The loss was determined by comparison between average gene number per subgenome against that of diploid *E. haploclada*. (34272, 34927, 31232 for *E. oryziicola*, *E. crus-galli* and *E. colona*, respectively, vs 37510 for *E. haploclada*).

Line 220: Barabaschi et al. 2020 posits a fitness cost but makes no such calculation. Rather these authors infer it based on genomic comparisons. Similarly, Ye et al. 2020 do not seem to measure fitness cost but rather gene expansion. Recommend that authors cite Review of fitness cost by Brown and Rant 2013, or other similar sources used in Ye et al. 2020 to first establish the strong link between R genes and fitness cost.

Response:

We have changed the citation.

Line 228 and 230: “significantly” is used again without obvious statistical method

Response:

We removed it in line 228 and added statistic test in line 230

Lines 303-323: The authors set out to establish if discordance between predicted discordance times and ks estimates are driven by incomplete lineage sorting and/or introgression. The authors calculated ks divergence using a COGE like approach, if not COGE exactly. Phylogenetic methods, were based off of Orthofinder and RAxML. Coalescent approach done with ASTRAL. While the authors work using QuIBL is respectable, their handling of the concatenated RAxML tree is unsatisfying. They report no alignment QC, partitioning or model selection. While Orthofinder is a nice general purpose program for generating Orthogroups, it is no substitute for handling the data rigorously. Questions abound.

Response:

Thanks for your comment. We have updated the phylogenetic analyses under strict

alignment QC and clarified the model selection clearly in Methods. Please also see responses for the following questions.

1. Why not use nucleotide sequences with a codon aware aligner instead of the proteins. If the species are close enough to calculate reliable ks peaks, aligning at the nucleotide level is a tractable task.

Response:

We used the CDS sequences of single-copy orthologs to construct concatenation trees. The phylogeny trees based on Raxml and iqtree+ASTRAL using PROT and CDS datasets are consistent.

Phylogenetic trees using RaxML and iqtree+ASTRAL for amino acid sequences (PROT) and nucleotide sequences (CDS) datasets

2. Since whole genome sequences are being used, why not employ the same set of homologous genes identified via synteny in genome guided orthology approach? This would be even more reliable than orthofinder since it incorporates genomic position, and basic data handling would allow for the use of amino acid or nucleotide sequences as authors saw fit.

Response:

Thanks for your suggestions. It is an ideal approach for constructing species phylogeny based on syntenic orthologs. While it is still a challenge to do multi-genome alignment due to the widespread genome-wide structural variations (e.g. gene loss and duplication). Here we employed a frequently-used method to build the species phylogeny by using 3500+ single-copy orthologs. The gene number is sufficient to sort the phylogenetic relationship.

In the case where Orthofinder is employed. I caution the authors that past experience with Orthofinder strongly suggests that the data will only benefit from a more rigorous handling of the sequences. This is because the alignments are frequently gappy and poorly aligned under alignment settings provided by most automated aligners and thus a QC step is required. The QC step and post processing should address the following.

1. How were gaps dealt with, the introduction of gaps into multiple sequence alignments can have a severe consequences.

Response:

GBlocks (v 0.91b) was used to remove gaps (parameter -b4=5, -b5=h).

2. Was GBlocks or a similar program employed to limit the amount of noise in the alignment? If so why was it not reported and what were the flags used to ensure reliable alignments were used in the concatenated sequence

Response:

We have updated the trees using alignment after trimming by GBlocks (-b4=5 -b5=h).

3. Was partitionfinder2 or some suitable substitute used to ensure that the correct models were applied throughout the concatenated matrix?

Response:

We used jModelTest and ProtTest to test the best substitution model for nucleotide and amino acid sequences. And also the best models tested by ModelFinder implemented in IQ-TREE were also considered.

4. What model was used for RAxML?

Response:

For amino acid (PROT) dataset, the substitution model was set as PROTGAMMA+JTT+F, as tested by ProtTest and ModelFinder in IQ-TREE. For nucleotide (CDS) dataset, the model GTRGAMMA was selected, as tested by jModelTest and ModelFinder in IQ-TREE.

The same point about alignment QC applies to the Multiple Sequence Alignments (MSAs) used for IQTree. IQtree does an excellent job testing for best substitution models, however it does not manage gapped or poorly aligned sequenced output

regularly produced by MAFFT with default settings. This is particularly problematic for sequences with large disordered domains and for orthogroups in which one or more sequences contains novel amino acid sequence insertions. These must be dealt with before MSAs are given to IQTree. Again, there is no obvious evidence that this was done. Poor alignments produce unreliable inferences.

Response:

Thanks for your comment. We have updated the phylogenetic analysis by IQ-TREE using GBlocks-trimmed alignments (-b4=5 -b5=h). The species phylogeny and triplet phylogeny results are consistent with previous results.

Species phylogenetic trees by IQ-TREE and ASTRAL using datasets of amino acid sequences (PROT, left) and nucleotide coding sequences (CDS, right)

The authors have not convinced me that they have handled the analysis rigorously enough to support the claims based on this analysis. In particular, it seems just as likely that they are measuring artifact introduced by the handling of data as they are measuring ILS or introgression. I admit that measuring ILS and introgression is a worthy undertaking, but has not been convinced by the current handling of the data.

Response:

Thanks for your comments. Yes, it is not easy to distinguish introgression and ILS. Here, we considered the radiation of *Echinochloa* at the first divergence node as an ILS event, based on short interval length and low bootstrap value from species phylogenies (Supplementary Figs. 13a and 13b), balanced phylogeny proportion in triplet (*O. sativa*, *E. crus-galli* AH, *E. colona* FH, *E. haploclada*) (Supplementary Fig. 13c), and QuIBL test, a method used in quantifying the contribution of ILS or introgression to discordance of gene trees against species tree (Supplementary Table 2). QuIBL estimates the distribution of internal branch length in discordant topologies for triplets of species, and then calculates the likelihood that this distribution corresponds to the model with introgression and ILS or with ILS only. QuIBL was successfully employed in explaining the radiative evolution of butterfly (Edelman et al., 2019, Science, 10.1126/science.aaw2090), Peatmoss (*Sphagnum*) (Meleshko et al., 2021, Mol. Biol.

Evol., doi:10.1093/molbev/msab063), Eared Seal (Lopes et al., 2020, Syst. Biol., 10.1093/sysbio/syaa099). In *Echinochloa*, QuIBL strongly indicated the main factor behind phylogenetic discordance among the *Echinochloa* subgenomes for radiation was ILS, rather than introgression.

Along with evolution, the footprints of ILS and introgression should be different. Although recombination would split the introgressed blocks, genes with the same discordant topology caused by introgression would be expected to be located together. In the evolution of *Heliconius* butterfly, large blocks showing the same topology were observed (Edelman et al., 2019, Science, 10.1126/science.aaw2090).

Fig. 2. Local evolutionary history in the erato-sara clade is heterogeneous across the genome. (A) Each bar represents a chromosome, in terms of the *H. erato* reference (14). Colored bands represent tree topologies of each 50-kb window; colors correspond to the topologies in (B), with black regions showing missing data. **(B)** The eight most common trees. The value in the top left corner is the percentage of all 50-kb windows that recover that topology. **(C)** Each histogram corresponds to the topology of the same color in (B) and shows the distribution of the number of consecutive 50-kb windows with that topology. Arrows indicate long blocks in inversions.

Fig. 2 in Edelman et al., 2019, Science, 10.1126/science.aaw2090

While in *Echinochloa*, we observed symmetric and unbiased distribution among three topologies in triplet (*O. sativa*, *E. crus-galli* AH, *E. colona* FH, *E. haploclada*) along nine chromosomes of *E. haploclada*, which was in accordance with the pattern of ILS.

Distribution of 3,557 single-copy genes on nine chromosomes of *E. haploclada*, used in phylogenetic analysis. Dots with three different colors represent three topology trees, respectively

Ln574: Please cite “Tranel, Patrick J., and Terry R. Wright. "Resistance of weeds to ALS-inhibiting herbicides: what have we learned?." *Weed Science* 50, no. 6 (2002): 700-712.” or “Murphy, Brent P., and Patrick J. Tranel. "Target-site mutations conferring herbicide resistance." *Plants* 8, no. 10 (2019): 382.” It is an essential piece of literature addressing TSR for ALS inhibitors

Response:

We have added the literature.

Line 613: Please cite “Patterson, Eric L., Dean J. Pettinga, Karl Ravet, Paul Neve, and Todd A. Gaines. "Glyphosate resistance and EPSPS gene duplication: convergent evolution in multiple plant species." *Journal of Heredity* 109, no. 2 (2018): 117-125.” It is a more focused review of the effect that CNVs have on glyphosate resistance.

Response:

We have added the literature.

Lines: 668-675 “Significantly” is used twice without an obvious statistical method. The pipeline CNVings is briefly described, but no mention is made of statistical test is applied to determine differences, perhaps it was a T-test.

Response:

Thanks for your comment. We have added the statistical methods for “significant”.

Figure 1:

Line 1106-1112: This circus plot can be deconvoluted. Many of these tracks are unnecessary (e.g. GC content) as they don't increase our understanding of the narrative and are for the most part uninteresting (especially crammed in a tiny figure). Only include the data relevant to your point(s). This will highlight the more relevant tracks (like Copia, Gypsy, etc).

Response:

Thanks for your suggestion. We have deleted the track about the GC content.

Line 1113: Not sure this is interesting in a graphical form. Just extra complexity and we understand well from the text how complete the genome(s) are. Also, it may allow more room for panel D and/or E to be larger and more interpretable.

Response:

We have removed this sub-figure.

Figure 2:

Line 1139-1141: The size of this figure versus the complexity of the figure are preposterous. A simple cartoon, collapsing redundant branches, would suffice to tell the admixture story your are trying to tell.

Response:

Here we expanded all the branches aiming to show the population structure and phylogeny of *Echinochloa* species/subspecies more intuitively.

Figure 3:

Line 1161-1164: Same comment as before, what benefit do we get as a reader to seeing every branch and branch length? Simplify your figures where you can to tell a more concise and interpretable story.

Response:

Here we expanded all the branches to show the population structure and phylogeny of *Echinochloa* species/subspecies more concretely based on chloroplast genomes, and to show more clearly the phylogenetic positions of outliers as indicated by red asterisks.

Line?: The figure panel of the two clades (Below the ML phylogeny) is not described in the figure head.

Response:

Thanks for your comment. We have added the legend.

Figure 5: Why are panel 1D separated so far from Figure 5 when both are discussing NB-ARC? Shouldn't panel 1D be part of this figure instead?

Response:

Figure 1D illustrated the NB-ARC number base on single reference genomes aiming to explore the dynamics of copy number among species, while Figure 5 discussed the copy number alteration by domestication (estimated by NGS resequencing data).

Reviewer #4 (Remarks to the Author):

The article brings novel insights into an evolution and agronomic significance of high polyploid grasses. Although there are several genome assemblies of allopolyploid species already available, the comprehensive analysis presented here brings valuable general insights into allopolyploid evolution in general. The model system is fascinating and in depth investigated by an impressive collection of global diversity of the three focal species. The analyses are diverse and at many aspects very comprehensive (esp. genome assembly, investigation of range-wide diversity) while the analysis of selective footprints of local adaptation, herbicide resistance and domestication is rather limited to testing particular hypotheses, which are addressed by a diverse and somehow inconsistent approaches, that seem not to be always clearly justified. Although I do not see any major significant flaw, I have the following comments aiming to clarify and potentially improve some aspects of the study.

Response:

Thanks for the very positive comments on our manuscript.

1) The phylogeny of *Echinochloa* species and subgenomes is crucial for further interpretations. I understand the reasons why there are no more genomes assembled and sequences but I wonder whether there are at least some sequence data for additional species available. Addition of such data (e.g. in a supplementary tree involving a subset of loci gathered also from current genomes) might better clarify the origin of the other subgenomes, esp. FH and EH that are currently isolated in the tree. Putting these newly assembled (sub)genomes into a broader context of *Echinochloa* evolution would help understanding their evolution.

Response:

Thanks for your comments. Currently *E. crus-galli*, *E. colona* and *E. oryzicola* are

the dominant barnyardgrass species. No other *Echinochloa* genomic resources could be used, besides the data in this study. To better classify the origin and evolution of *Echinochloa*, obtaining genomes of more *Echinochloa* species including diploids is necessary, which is what we are about to do.

2) Many recently assembled allopolyploid genomes demonstrate surprising stasis and lack of large genomic exchanges among the sub-genomes (e.g. Burns et al. 2021 Nat Ecol Evol and refs therein), however, it would be still worth checking and discussing if that is also the case for *Echinochloa* or whether they are in fact such homeologous recombinations present. Lack of homeologous exchange is an assumption of many downstream analyses but has not been discussed.

Response:

Thanks for your comment.

For *E. crus-galli*, in the process of genome assembly, we have noticed the existence of HEs in its genome at the step of contig correction (Supplementary Note 4). The parental genomes are available, which provide the chance to comprehensively profile the HEs in *E. crus-galli*. By mapping ancestral genomes, we identified 24 candidate HEs in *E. crus-galli* (see the following figure). These candidate HE events were supported by HiC interaction and some are supported by the mapping mate-paired reads, which were unlikely to be caused by mis-assembly (see the following figures).

HEs would interfere the phylogeny based on whole-genome genes and genes within HE regions should be removed before phylogeny construction. Actually, none of the previously identified 3,557 single-copy genes were located within the candidate HE regions. Large HEs were observed to be close with telomeres (chromosome ends) (e.g. HEs on BH01, CH01, BH05) where genes are poor. Thus, the HEs would not influence the conclusions of phylogenetic inference in our case.

Distribution of candidate HEs (left) and mapping depth on 27 chromosomes of *E. crus-galli* (right)

Blue, green and red dots represent the depth when genomic sequences (50× depth) from subgenome AT, BT from *E. oryziicola* and *E. haploclada* were mapped against *E. crus-galli* genome, respectively. Candidate HEs are marked by gray rectangles. Mapping around numbered HEs using 10kb-insert mate-paired reads are shown in IGV.

HiC interaction at single-chromosome scale in *E. crus-galli* genome
 None of abnormal signals are observed around HE regions (e.g. terminal regions on chromosome BH01, CH01, BH05).

Due to the lack of direct ancestors of *E. oryzicola* and *E. colona*, we did not check the HEs in these two species.

3) What is the breeding system of the species? In case of strong selfing, this shall be taken into account in the interpretations of diversity and differentiation from population resequenced data. E.g. could the decrease in nucleotide diversity among *E.c.* varieties (l. 650-653) be attributed to breeding system shift, possibly associated with domestication?

Response:

Yes, the breeding system of *Echinochloa* is selfing. As we know and observe, no breeding system shift occurs among *E. crus-galli* varieties.

4) 1 217-240 The finding of contraction of NB-ARC gene family is interesting, it is however unclear how much this case represents an outlier in gene contraction among different gene families in *Echinochloa*. Hence, it is hard to speculate on adaptive significance of such a reduction (see l. 219) and it is a bit unclear the motivation why an entire paragraph is devoted to discussion of this particular family. Is the NRC-ARC family contraction significantly higher than is a genome-wide average in *Echinochloa*?

Response:

Thanks for your comments. A focus on the NB-ARC gene family was not a random selection, rather it was based on our analysis results on the dynamics of family sizes of several genes with a role in abiotic and/or biotic stress response (see Supplementary Fig. 8 and the following Table) and that the fact of NB-ARC being crucial in plant's fitness. As you can see from the following Table, compared to abiotic response gene families, biotic response gene families in *Echinochloa* showed relatively smaller in size,

compared to its neighboring species, such as *Alloteropsis semialata* and *Setaria italica*, particularly the NB-ARC family.

Comparison of abiotic and biotic gene family sizes

Species	Total gene counts	NB-ARC	Legume lectin	D-mannose binding lectin	BTB/POZ	UDP	AP2	C2	GRAS	MYB	ABC transporter	CYP
S. italica	34584	368	60	122	115	211	136	69	57	149	134	359
A. semialata	45144	585	60	92	168	200	113	64	56	140	139	421
E. oryzicola AT	33635	124	13	59	75	64	100	62	52	141	117	309
E. crus-galli AH	33261	164	12	44	27	40	109	60	30	140	109	250
E. colona FH	29510	142	10	37	82	47	112	64	49	140	116	226
E. oryzicola BT	33316	134	16	48	104	78	112	60	48	138	125	322
E. crus-galli BH	34124	152	24	49	34	53	107	62	32	146	125	299
E. colona EH	30593	140	20	43	130	61	105	64	47	144	120	227
E. crus-galli CH	35808	186	18	44	26	60	122	61	38	152	124	300
E. colona DH	33538	165	12	31	121	56	115	60	50	141	100	239
E. haploclada	37510	260	17	51	27	59	125	68	37	160	135	283
Echinochloa Average	33477	163	16	45	70	58	112	62	43	145	119	273
P1(Echinochloa vs S. italica)		8.63E-18	7.31E-07	5.80E-09	2.44E-03	1.50E-20	0.227	0.727	0.231	1	0.529	0.003
P2(Echinochloa vs A. semialata)	fisher's exact test	1.30E-32	1.00E-04	2.40E-02	3.26E-05	1.84E-11	0.031	0.149	0.919	0.006	0.257	0.09

5) Fig. 1E: It is totally unclear from where the time estimates come from. Are they based on some uniform substitution rate estimate in grasses? If so, such an assumption seem to be tricky given known variation in substitution rates, and surely presenting a single point estimate of divergence time, based on such an approach, may be very misleading. It will be beneficial either to date the splits based on calibrated phylogeny (including confidence intervals for each node), or such numbers shall be excluded from the main figure to avoid potential misleading divergence times (these values seem not to be very important for further interpretation anyway). Related to that, a phylogeny based on multispecies coalescent reconstruction across multiple loci (i.e. Astral, Fig S13), i.e. an approach directly developed for phylogeny reconstructions, may provide more robust than the relatedness based on the distribution of nonsynonymous substitutions.

Response:

Thanks for your suggestions. We have rebuilt this sub-fig by removing the divergence time and replacing *Ks*-based phylogenetic tree with multi-loci coalescent tree by ASTRAL.

6) The motivation for using particular methods for detection selection is not always clear and raises concerns about the homogeneity of the approaches throughout the ms. Specifically, why results of flowering-time variation are based on a genome-wide divergence (*Fst*) outlier scan while e.g. the domestication process is investigated by allele frequency spectra-oriented approach targeting only a subset of (pre-selected?) loci?

Response:

Here we focus on domestication-related genes that have been cloned in rice. The basic molecular biology studies are very limited for *Echinochloa*, thus in order to have an overall understanding to *Echinochloa* domestication, the cloned domestication-related genes in rice provide good resources to explore the genetic mechanism behind the barnyard millet domestication. The analyses of domestication-related genes in barnyard millet is straight-forward, and the haplotype profiling in *Echinochloa* is comparable against rice domestication. Genome-wide scan (e.g. *Fst*) to detect domestication regions is popular and useful, but the scan results usually contain a large proportion of false positive genes with unknown function and roles in domestication. Nevertheless, genome-wide scan of differentiation between wild and cultivated *Echinochloa* will provide valuable resources for following molecular studies and millet breeding and improvement. We have added genome-wide *Fst* scan and listed all the genes within the top 5% genomic regions with cloned rice genes annotation in Supplementary Tables 15-17 in this revise version.

7) In some candidate loci it is unclear how the variation with these loci looks in the other two homeologs within a hexaploid (e.g. *Hd1* is reported only from subgenome AH but nothing is mentioned about subgenome BH and CH). Are they pseudogenes? Invariable but likely functional loci? Variable but not enough to be considered outliers? Undetected? Such a context would provide important hints about the role of allopolyploidisation and potential sub/neofunctionalisation in adaptation.

Response:

Thanks for your suggestion. We have added the analyses on differentiation and selection of other two copies of *Hd1* in subgenomes BH and CH (Supplementary Figure 31). Although *Hd1* in subgenome AH (AH06.1150) showed highly differentiation (greater than top 1% threshold genome-wide) between *E. crus-galli* var. *crus-galli* and var. *praticola* and selection signal in var. *crus-galli*, no obvious differentiation and selection (less than thresholds) was observed for *Hd1* in subgenome BH and CH (BH06.1184 and CH06.1302), which suggested preferential differentiation and selection in polyploid *Echinochloa* species. Also in *E. oryzicola*, *Hd1* on chromosome AT06 showed differentiation and selection for low-latitude group, while no signal was shown for *Hd1* on chromosome BT06. Related results have been added in the main text.

Preferential differentiation and selection on *Hd1* in *E. crus-galli* and *E. oryzicola*

(a) genomic differentiation scan in 20-kb sliding windows around three orthologs of *Hd1* (AH06.1150, BH06.1184 and CH06.1302) on chromosome AT06, BT06 and CH06. Red dots represent the window harboring *Hd1* genes. Red and black dashed lines represent thresholds of top 1% and 5% F_{st} values, respectively. (b) Tajima's D scan in 20-kb sliding windows around *Hd1*. Red dashed lines represent 95% significance thresholds. (c) nucleotide diversity π in 20-kb sliding windows around *Hd1*.

8) The entire section on herbicide resistance (l. 569-627) lacks a primary exploratory genome-wide approach and/or justification why the entire parts focuses only on a small subset of particular loci. As a result, it appears as somehow selective "cherry-picking" of certain loci without a justification how much variation is such loci is exceptional in the context of the entire genome. Thus, although this section provides some hints of potentially valuable loci for a follow-up it does not adhere to the major focus of the paper on genome-wide variation and evolutionary significance

of genomic variation related to allopolyploidy. At current state this section provides rather some particular supplementary information targeted to a very specialised audience.

Response:

Thanks for your comments. As weeds, world-wide *Echinochloa* grass suffered from not only local natural selection, but also artificial stress by agricultural managements. Human-directed recent selection pressure on weeds largely comes from the wide application of herbicides. You are right that the loci presented in the section were not picked up by a genome-wide scan as that requires functional evidence (can be generated by large scale phenotyping and GWAS analyses following by functional confirmation) of all possible loci associated with herbicide resistance, which is current unavailable. We thus focused on the genes with a confirmed role in herbicide resistance and surveyed their variations in the world-wide *Echinochloa* accessions to generate a mutation atlas of both target site (TSR) and non-target-site resistance (NTSR), as we believe that the variations contributed to herbicide resistance are very important for evolution of *Echinochloa* species and their adaptation to diverse or localized agrosystems.

While the trait of herbicide resistance per se may be not of interest for those working on polyploid and crop evolution, it's a very important and demanding topic of agriculture as a whole. Thus, the results should be of general interest.

9) The Discussion is very brief and descriptive. Although not being an expert in the filed of agricultural genomics, I doubt there is so scarce literature on the topics as it is presented here (in total four citations in the entire Discussion section). E.g. considering additional well-researched cases of the evolution of agricultural weeds such as *Amaranthus tuberculatus* (Kreiner et al. 2019 PNAS, Kreiner et al. 2018 ARPB and refs. therein) would be helpful. Also putting the result into a context of other emerging studies on genomic evolution in allopolyploids (e.g. *Arabidopsis suecica*, *Brassica napus*, *Gossypium*, ...) might increase the impact of the study.

Response:

Thanks for your suggestions. We have overhauled the discussion section by discussing a bit deeper on the relevant topics.

10) Although the authors at multiple places highlight the significance of their study for follow-up genetic studies, list of candidate loci for adaptation towards the discussed factors (resistance, latitude, domestication...) are missing. It would be evry helpful providing complete lists of candidate loci found genome wide by Fst scans,

GWAS approaches and SFS-based scans as such primary data would provide molecular biologist a key resource for addressing particular genes in this emerging model.

Response:

Thanks for your suggestion. We have listed all genome scan results and candidate genes in the supplementary materials in the revised version of the MS.

Minors

- I consider crucial the fact the studied group is ALLOpolyploid. I suggest adding this information already to the abstract.

Response:

We have added the information in the abstract.

- l. 105-106 maybe "most" missing from "the MOST compact genome"

Response:

We have revised it.

- l. 253 "calculated the pairwise Ks" is a jargon. Please explain the principle/motivation behind this analysis.

Response:

We have updated the wording.

- l. 321 based on what the authors conclude that "introgression may have facilitated an increase in environmental adaptation"? This seems rather speculative given the presented Ks analysis.

Response:

We have deleted this sentence.

- l. 338 what is the ploidy of *E. walteri* that is used as an outgroup in the resequencing data?

Response:

E. walteri is tetraploid.

- l. 397 unclear, which measure the % of "admixture" are based on

Response:

E. walteri is tetraploid. The % of "admixture" were estimated by fastsimcoal2.

- l. 506 unclear what are the "outliers", this shall be also explained in the text not only in the Fig caption

Response:

The outliers are *Echinochloa* individuals with abnormal genome mapping coverage and read mapping rate or odd phylogenetic positions in phylogenetic tree (Supplementary Table 4). We have improved the wording here.

- l. 536 how many individuals were used for the GWAS? How many loci others than Hd1 were found to be significantly associated with FT?

Response:

A total of 461 individuals of *E. crus-galli* were used for the GWAS. All other loci associated with flowering time are listed in Supplementary Table 10.

Reviewers' Comments:

Reviewer #1:

Remarks to the Author:

The authors present a substantially revised version of the manuscript and addressed most of my queries. However, I am still not completely satisfied with the use of the term 'chromosome-scale'. I am aware that this term is not used consistently in literature. In my opinion, however, there needs to be a clear distinction between assemblies that are generated completely de novo (i.e., only use sequencing and scaffolding approaches to reach chromosome scale) and assemblies that are generated by using comparisons to other species (in other words are not completely de novo and independent). DipHiC 'integrates the genetic relationship to diploid species' and is thus not completely independent and de novo. I do not question the quality of the Echinochloa colona assembly and I appreciate the challenges to assemble polyploid genomes with highly similar sub-genomes, but it is important to clearly make the above distinction in order to establish clarity. The statement 'although we believe that the assembly could be further improved by integrating information from other approaches' is true for most of the genome assemblies. This might be a minor point, but it is important in my view and the authors should address it before publication of this manuscript.

Reviewer #2:

Remarks to the Author:

According to the Response Letter and the revised manuscript, I can understand the authors have do their best to address reviewers' concerns and improve the manuscript.

For my major concerns in previous review :

- 1) The necessary of the application of DipHiC in allopolyploid genomes. In Response Letter, authors have provided more details, which convinced me that DipHiC can be useful on some complex genomes that present Hic tools fail to assemble.
- 2) The organization and wording of manuscript have been refined according to the comments from four reviewers.

Some minor comments:

- 1) The accuracy of subgenome phase was validated by the subgenome-specific kmer. In Response Letter, authors explained the K-mers, but not the method to use the K-mers. From the supplementary figure 2, I saw the K-mers are used for clustering. The clustering method needed to be explained in the METHOD and relative references needed to be added.
- 2) I am wondering whether 'phasing' can be used to describe the 'distinguish' of subgenomes. It may be confused with the assembly of homologous haplotypes in heterozygous genome.

Overall, I suggest the manuscript meet the requirement for publication after some minor improvement.

Reviewer #3:

Remarks to the Author:

I thank the authors for their thoughtful and thorough responses to the reviewers. I believe the manuscript has been made significantly better with the changes and I have no further comments that will significantly increase the quality of this manuscript. I look forward to seeing the manuscript in print!

Reviewer #4:

Remarks to the Author:

Major points

1) In general I appreciate the answers in the rebuttal letter which well addressed my questions, however, some of the key points remained not reflected in the main text. As I likely would not be the only reader having such questions I recommend incorporating of these issues directly into the text, briefly. In particular:

-Mentioning that other (unsampled here) diploid *Echinochloa* genomes may be the parents.

-adding the information that none of the previously identified 3,557 single-copy genes were located within the candidate HE regions to the Methods, this will remove potential concerns on the phylogeny reconstruction. NB this could also mean that there is negative selection against HEs, if they only exist in gene poor regions?

-Explaining that *Echinochloa* is selfing (with refs) and how the fact that selfing is widespread in this genus can explain the demography of the species, esp. historical var. *crus-galli* population contraction (l. 945)..

-*E. walteri* – it would be useful to add the information about tetraploidy of this species to the main ms

2) If the Fig 1E has been replaced by Astral analysis as stated in the rebuttal letter (#4, comment 5), the Figure caption still refers to a Ks analysis - this shall be corrected.

3) The genome assemblies seem to be currently unavailable. E.g. under the link of *E. crus galii* chromosome-scale assembly (<https://ngdc.cnbc.ac.cn/biosample/browse/SAMC393329>) there is no "Sample Data" available for download. If this is just a matter of data embargo, please explain.

Minors

| 951 replacement of barnyard millets by cold-tolerant rice varieties?

Fig 1d please use full genus names outside *Echinochloa*. It is unclear for a non-grass specialist which genera are included there.

REVIEWERS' COMMENTS

Reviewer #1 (Remarks to the Author):

The authors present a substantially revised version of the manuscript and addressed most of my queries. However, I am still not completely satisfied with the use of the term 'chromosome-scale'. I am aware that this term is not used consistently in literature. In my opinion, however, there needs to be a clear distinction between assemblies that are generated completely de novo (i.e., only use sequencing and scaffolding approaches to reach chromosome scale) and assemblies that are generated by using comparisons to other species (in other words are not completely de novo and independent). DipHiC 'integrates the genetic relationship to diploid species' and is thus not completely independent and de novo. I do not question the quality of the *Echinochloa colona* assembly and I appreciate the challenges to assemble polyploid genomes with highly similar sub-genomes, but it is important to clearly make the above distinction in order to establish clarity. The statement 'although we believe that the assembly could be further improved by integrating information from other approaches' is true for most of the genome assemblies. This might be a minor point, but it is important in my view and the authors should address it before publication of this manuscript.

Response: Thanks. We have deleted the term "chromosome-scale" and changed it to be "high quality"

Reviewer #2 (Remarks to the Author):

According to the Response Letter and the revised manuscript, I can understand the authors have do their best to address reviewers' concerns and improve the manuscript.

For my major concerns in previous review :

1) The necessary of the application of DipHiC in allopolyploid genomes. In Response Letter, authors have provided more details, which convinced me that DipHiC can be useful on some complex genomes that present Hic tools fail to assemble.

2) The organization and wording of manuscript have been refined according to the comments from four reviewers.

Response: Thanks.

Some minor comments:

1) The accuracy of subgenome phase was validated by the subgenome-specific kmer. In Response Letter, authors explained the K-mers, but not the method to use the K-mers. From the supplementary figure 2, I saw the K-mers are used for clustering. The clustering method needed to be explained in the METHOD and relative references needed to be added.

Response: Thanks for your suggestion. We have added K-mer analysis in Method.

2) I am wondering whether 'phasing' can be used to describe the 'distinguish' of subgenomes. It may be confused with the assembly of homologous haplotypes in heterozygous genome.

Response: We reword "phase" by "distinguish".

Overall, I suggest the manuscript meet the requirement for publication after some minor improvement.

Response: Thanks.

Reviewer #3 (Remarks to the Author):

I thank the authors for their thoughtful and thorough responses to the reviewers. I believe the manuscript has been made significantly better with the changes and I have no further comments that will significantly increase the quality of this manuscript. I look forward to seeing the manuscript in print!

Response: Thanks.

Reviewer #4 (Remarks to the Author):

Major points

1) In general I appreciate the answers in the rebuttal letter which well addressed my questions, however, some of the key points remained not reflected in the main text. As I likely would not be the only reader having such questions I recommend incorporating of these issues directly into the text, briefly. In particular:

-Mentioning that other (unsampled here) diploid *Echinochloa* genomes may be the parents.

Response: We have added the statement pointing out that *E. haploclada* is a closely related species of maternal donor of *E. crus-galli* in Introduction section.

-adding the information that none of the previously identified 3,557 single-copy genes were located within the candidate HE regions to the Methods, this will remove potential concerns on the phylogeny reconstruction. NB this could also mean that there is negative selection against HEs, if they only exist in gene poor regions?

Response: Thanks for your suggestion. We have added the statement about HE in Method part.

In our case, it seems to be that there might be negative selection against HEs, however, it needs more data to make conclusion.

-Explaining that *Echinochloa* is selfing (with refs) and how the fact that selfing is widespread in this genus can explain the demography of the species, esp. historical var. *crus-galli* population contraction (l. 945)..

Response: Thanks for your suggestion. *E. crus-galli* plants are self-compatible and highly autogamous (Maun and Barrett, 1986). In our case, we investigated recent demographic history within the last 10 thousand years and found population contraction at several thousand/hundred years ago, e.g., 400 years ago for var. *crus-galli*. Our reasonable explanation is that farming practices caused contraction of *E. crus-galli* population, e.g., rice transplanting (instead of direct seeding) for var. *crus-galli*. We agree with you that selfing reduces the effective population size,

however, it may not be caused by selfing for *Echinochloa*, as no evidence shows that mating system shifts or selfing rate increases during the varieties differentiation and domestication.

References: Maun, M. A. & Barrett, S. C. H. The biology of Canadian weeds. 77. *Echinochloa crus-galli* (L) Beauv. Can. J. Plant Sci. 66, 739-759 (1986).

-E. walteri – it would be useful to add the information about tetraploidy of this species to the main text

Response: Thanks for your suggestion. We have added it.

2) If the Fig 1E has been replaced by Astral analysis as stated in the rebuttal letter (#4, comment 5), the Figure caption still refers to a Ks analysis - this shall be corrected.

Response: We are sorry about it. We have corrected it.

3) The genome assemblies seem to be currently unavailable. E.g. under the link of *E. crus-galii* chromosome-scale assembly (<https://ngdc.cncb.ac.cn/biosample/browse/SAMC393329>) there is no "Sample Data" available for download. If this is just a matter of data embargo, please explain.

Response: All genome assemblies and annotations, and resequencing data are released currently and are available at NGDC BioProject PRJCA003883 (<https://ngdc.cncb.ac.cn/search/?dbId=&q=PRJCA003883>).

Minors

1951 replacement of barnyard millets by cold-tolerant rice varieties?

Response: We have reworded it.

Fig 1d please use full genus names outside *Echinochloa*. It is unclear for a non-grass specialist which genera are included there.

Response: We have changed as suggested.